# Lower Bounds and Nearly Optimal Algorithms in Distributed Learning with Communication Compression

**Xinmeng Huang**[1*]      **Yiming Chen**[2,3*]      **Wotao Yin**[2]      **Kun Yuan**[2,4]

[1]University of Pennsylvania   [2]DAMO Academy, Alibaba Group
[3]MetaCarbon   [4]Peking University

xinmengh@sas.upenn.edu   yiming@metacarbon.vip
{wotao.yin, kun.yuan}@alibaba-inc.com

## Abstract

Recent advances in distributed optimization and learning have shown that communication compression is one of the most effective means of reducing communication. While there have been many results for convergence rates with compressed communication, a lower bound is still missing.

Analyses of algorithms with communication compression have identified two abstract properties that guarantee convergence: the unbiased property or the contractive property. They can be applied either unidirectionally (compressing messages from worker to server) or bidirectionally. In the smooth and non-convex stochastic regime, this paper establishes a lower bound for distributed algorithms whether using unbiased or contractive compressors in unidirection or bidirection. To close the gap between this lower bound and the best existing upper bound, we further propose an algorithm, NEOLITHIC, that almost reaches our lower bound (except for a logarithm factor) under mild conditions. Our results also show that using contractive compressors in bidirection can yield iterative methods that converge as fast as those using unbiased compressors unidirectionally. We report experimental results that validate our findings.

## 1 Introduction

Large-scale optimization is a critical step in many machine learning applications. Millions or even billions of data samples contribute to the excellent performance in tasks such as robotics, computer vision, natural language processing, healthcare, and so on. However, such a scale of data samples and model parameters leads to enormous communication that hampers the scalability of distributed machine-learning training systems. We urgently need communication-reduction strategies. State-of-the-art strategies include model and gradient compression [4, 12, 56], decentralized communication [39, 9, 19], lazy communication [73, 57, 33, 18], and beyond. This article will focus on the former.

The most common method of distributed training is Parallel SGD (P-SGD) [15]. In P-SGD, the stochastic gradients that workers transmit to a server cause significant communication overhead in large-scale machine learning. To reduce this overhead, many recent works propose to compress the messages sent unidirectionally from worker to server [5, 28, 4] or compress the messages between them bidirectionally [62, 70]. The method of compression is either sparsification or quantization [5, 28, 4] or their combination [28, 14]. The literature [58, 55, 62, 70] reveals that bidirectional compression can save more communication, but leads to slower convergence rates.

---

*Equal Contribution. Corresponding Author: Kun Yuan

Table 1: Comparison between various distributed stochastic algorithms with communication compression for non-convex loss functions. To explicitly clarify the influence of different compression strategies, we keep the stochastic gradient variance $\sigma^2$, data heterogeneity bound $b^2$, gradient square norm bound $G^2$ (used in [58, 62, 68], much larger than $b^2$ and $L$), but omit smoothness constant $L$, and initialization $f(x^{(0)}) - f^\star$ in the below results. Moreover, notation $\tilde{O}(\cdot)$ hides all logarithmic terms. "LB" and "UB" indicate lower and upper bound, respectively. Notation $\delta$ and $\omega$ are compressor-related parameters (see detailed discussions in Section 2).

| | Algorithm | Convergence Rate | Compression[a] | Trans. Compl.[b] |
|---|---|---|---|---|
| **LB** | Theorem 1+Corollary 1 | $\Omega\left(\frac{\sigma}{\sqrt{nT}} + \frac{1}{\delta T}\right)$ | Uni/Bidirectional Contractive | $\Omega(n/\delta^2)$ |
| | Theorem 2+Corollary 2 | $\Omega\left(\frac{\sigma}{\sqrt{nT}} + \frac{1+\omega}{T}\right)$ | Uni/Bidirectional Unbiased | $\Omega\left(n(1+\omega)^2\right)$ |
| **UB** | Theorem 3 | $\tilde{O}\left(\frac{\sigma}{\sqrt{nT}} + \frac{1}{\delta T}\right)$ | Uni/Bidirectional Contractive | $\tilde{O}(n/\delta^2)$ |
| | Corollary 3 | $\tilde{O}\left(\frac{\sigma}{\sqrt{nT}} + \frac{1+\omega}{T}\right)$ | Uni/Bidirectional Unbiased | $\tilde{O}(n(1+\omega)^2)$ |
| | Q-SGD [32] | $O\left(\frac{(1+\omega)^{1/2}\sigma + \omega^{1/2}b}{\sqrt{nT}}\right)^{\dagger,\diamond}$ | Unidirectional i.i.d, Unbiased | − |
| | MEM-SGD [58] | $O\left(\frac{\sigma}{\sqrt{nT}} + \frac{G^{2/3}}{\delta^{2/3}T^{2/3}} + \frac{1}{T}\right)^{\diamond}$ | Unidirectional Contractive | $O(n^3/\delta^4)$ |
| | Double-Squeeze [62] | $O\left(\frac{1}{\sqrt{nT}} + \frac{G^{2/3}}{\delta^{4/3}T^{2/3}} + \frac{1}{T}\right)^{\diamond}$ | Unidirectional Contractive | $O(n^3/\delta^8)$ |
| | CSER [68] | $O\left(\frac{\sigma}{\sqrt{nT}} + \frac{G^{2/3}}{\delta^{2/3}T^{2/3}} + \frac{1}{T}\right)^{\diamond}$ | Unidirectional Contractive | $O(n^3/\delta^4)$ |
| | EF21-SGD [24] | $O\left(\frac{\sigma}{\sqrt{\delta^3 T}} + \frac{1}{\delta T}\right)^{*}$ | Unidirectional Contractive | − |

[a] This column indicates the type of the compressor and in what direction the compression is applied.

[b] This column indicates the transient complexity, *i.e.*, the number of gradient queries (or communication rounds) the algorithm has to experience before reaching the linear-speedup stage, *i.e.*, $O(\frac{\sigma}{\sqrt{nT}})$.

[†] This convergence rate is valid only for $T = \Omega(n(1+\frac{\omega}{n})^2)$. Since the rate $\frac{(1+\omega)^{1/2}\sigma + \omega^{1/2}b}{\sqrt{nT}}$ is always worse than $\frac{\sigma}{\sqrt{nT}}$, the transient complexity is not available.

[*] Since the convergence rate does not show linear-speedup, the transient complexity are not available.

[◇] The rates are either strengthened by relaxing the original restrictive assumptions or extended to the same setting as NEOLITHIC for a fair comparison, see more details in Appendix D.

Although there are many specific compression methods, their convergence analyses are mainly built on unbiased compressibility or contractive compressibility. The literature [14, 54, 71] summarizes these two properties and how they appear in the analyses. An unbiased compressor compresses a long vector $x$ into a short vector $C(x)$ and satisfies $\mathbb{E}[C(x)] = x$, i.e., no bias is introduced. The contractive compressor may introduce bias, but its compression introduces much less variance. We give their definitions below. Although the contractive compressor can empirically work better, the analysis of the unbiased compressor yields faster convergence due to unbiasedness [14, 44, 29, 26].

Despite the quick progress made in compression techniques and their convergence, we do not yet understand the limits of algorithms with communication compression. Since unbiased and contractive compressibilities are the two representative characteristics of various compressors, we use them to theorize two types of compression methods. For each type, we intend to answer:

*What is the optimal convergence rate that a distributed algorithm can achieve when using any of the compression methods of this type?*

Here, we assume that only unbiased compressibility or contractive compressibility can be used, not considering any additional special compressor design; after all, any special design in the literature has been heuristic, whose effectiveness can be explained at best and not proved or quantified. So, we further clarify our question: *Given a class of optimization problems (specified below) and a class of compression methods of the same type, if we choose the worst combination of them to defeat an algorithm, what will the convergence rate that the best-defending algorithm can reach?* To our knowledge, they are fundamental questions not addressed yet.

## 1.1 Main Results

This paper clarifies these open questions by providing lower bounds under the non-convex smooth stochastic optimization setting, and developing effective algorithms that match the lower bounds up to logarithm factors. In particular, our contributions are:

- We establish convergence lower bounds for distributed algorithms with communication compression in the stochastic non-convex regime. Our lower bounds apply to any algorithm conducting unidirectional or bidirectional compression and using unbiased or contractive compressors. We find a clear gap between the established lower bounds and the existing convergence rates.

- We propose a novel **ne**arly **o**ptimal a**lgorith**m w**i**th **c**ompression (NEOLITHIC) to fill in this gap. NEOLITHIC can adopt either unidirectional or bidirectional compression, and is compatible with both unbiased and contractive compressors. Using any combination, NEOLITHIC provably matches the above lower bound, under an additional mild assumption and up to logarithmic factors.

- The convergence results of NEOLITHIC also imply that, for non-convex problems, algorithms using biased contractive compressors bidirectionally can theoretically converge as fast as those with unbiased compressors used unidirectionally. Before out work, it is only established in [48] that, for convex problems and unbiased compressors, algorithms with bidirectional compression can match with the counterparts with unidirectional compression.

- We provide extensive experimental results to validate our theories.

All established results in this paper as well as convergence rates of existing state-of-the-art distributed algorithms with communication compression are listed in Table 1. The transient complexity, which measures how sensitive the algorithm is to the compression strategy (see Section 3), is also listed in the table. The smaller the transient complexity is, the faster the algorithm converges.

## 1.2 Related Works

**Distributed learning.**   Distributed learning has been increasingly useful in training large-scale machine learning models [21]. It typically follows a centralized or decentralized setup. Centralized approaches [2, 38], with P-SGD as the representative, require all workers to synchronize with a central server per iteration. Decentralized approaches, however, are based on partial averaging in which each worker only needs to synchronize with its immediate neighbors. Well-known decentralized algorithms include decentralized SGD [45, 77, 76, 39], D$^2$[61, 74], stochastic gradient tracking [69, 36, 3, 30], and their momentum variants [40, 75]. The lazy communication [73, 57, 18] is also utilized to reduce communication overheads in which workers conducts multiple local updates before sending messages.

**Communication compression.**   To alleviate the communication overhead in distributed learning, researchers have proposed two mainstream communication compression methods: quantization and sparsification. Quantization [32, 60, 80] is essentially an unbiased operator with random noise. For example, [56] develops Sign-SGD by using only 1 bit for each entry whose convergence is studied in [12, 13, 66]. Q-SGD [4] compresses each entry with more flexible bits and enables a trade-off between convergence rates and communication costs. Sparsification, on the other hand, amounts to a biased but contractive operator. [65, 58] propose to transmit a small number of entries, randomly or by taking the largest ones, to achieve sparsity in communication. The theories behind contractive compressors are limited to those in [64, 41, 59] due to analysis challenges, and they are established with assumptions such as bounded gradients [82, 34] or quadratic loss functions [67]. More discussions on unbiased and biased compressors can be found in [14, 54, 53]. Communication compression can also be combined with other communication-saving techniques such as decentralization [42, 81].

**Error compensation.**   The error compensation (feedback) mechanism is introduced by [56] to mitigate the error caused by 1-bit quantization. [67] studies SGD with error-compensated quantization for quadratic functions with convergence guarantees. [58] shows that error compensation can reduce quantization-incurred errors for strongly convex loss functions in the single-node setting. Error-compensated SGD is studied in [5] for non-convex loss functions with no establishment of an improved convergence rate. The algorithm EF21 [52] applies to the deterministic setting, using contractive compressors unidirectionally. It can converge under very mild assumptions. It has been recently extended to the stochastic setting [24] yet no linear-speedup rate is achieved.

**Lower bounds in optimization.** Lower bounds are well studied in convex optimization [23, 1, 22, 20] especially when there is no gradient noise [8, 78, 10, 6, 25]. In non-convex optimization, [16, 17] propose a zero-chain model and show a tight bound for first-order methods. By splitting the zero-chain model into multiple components, the authors of [83, 7] extend the analysis technique to finite sum and stochastic problems. Recently, [43, 76] also show the lower bounds in the decentralized stochastic setting based on a similar approach. For distributed learning with communication compression, a useful work [49] establishes a lower bound for unidirectional/bidirectional compression for strongly convex problems with fixed learning rates and unbiased compressors. There are limited studies on lower bounds for non-convex distributed learning with unbiased/contractive compressions.

## 2 Problem Setup

We consider standard distributed learning with $n$ parallel workers. The data samples in each worker $i$ follow a local distribution $D_i$, which can be heterogeneous among all workers. These workers, together with a central parameter server, collaborate to train a model by solving

$$\min_{x \in \mathbb{R}^d} \quad f(x) = \frac{1}{n} \sum_{i=1}^n f_i(x) \quad \text{with} \quad f_i(x) = \mathbb{E}_{\xi_i \sim D_i}[F(x; \xi_i)], \tag{1}$$

where $F(x; \xi)$ is the loss function evaluated at parameter $x$ with sample $\xi$. Since the objective $f$ can be non-convex, finding a global minimum of (1) is generally intractable. Therefore, we turn to seeking a model $\hat{x}$ with a small gradient magnitude in expectation, *i.e.*, $\mathbb{E}[\|\nabla f(\hat{x})\|^2]$. Next we introduce the setup under which we perform a convergence analysis.

### 2.1 Function Class

We let the function class $\mathcal{F}_{\Delta, L}$ denote the set of all functions satisfying Assumption 1 for any underlying dimension $d \in \mathbb{N}_+$ and a given initialization point $x^{(0)} \in \mathbb{R}^d$.

**Assumption 1 (Smoothness).** *We assume each local objective $f_i$ has $L$-Lipschitz gradient,* i.e.,

$$\|\nabla f_i(x) - \nabla f_i(y)\| \leq L\|x - y\|$$

*for any $i \in \{1, \ldots, n\}, x, y \in \mathbb{R}^d$, and $f(x^{(0)}) - \inf_{x \in \mathbb{R}^d} f(x) \leq \Delta$ with $f = \frac{1}{n} \sum_{i=1}^n f_i$.*

### 2.2 Gradient Oracle Class

We assume each worker $i$ has access to its local gradient $\nabla f_i(x)$ via a stochastic gradient oracle $O_i(x; \zeta_i)$ subject to randomness $\zeta_i$, *e.g.*, the mini-batch sampling $\zeta_i \triangleq \xi_i \sim D_i$. We further assume that the output $O_i(x, \zeta_i)$ is a time-independent and unbiased estimator of the full-batch gradient $\nabla f_i(x)$ with a bounded variance. Formally, we let the stochastic gradient oracle class $\mathcal{O}_{\sigma^2}$ denote the set of all oracles $O_i$ satisfying Assumption 2.

**Assumption 2 (Gradient noise).** *For any $x \in \mathbb{R}^d$ and $i \in \{1, \ldots, n\}$, the oracle $O_i$ satisfies*

$$\mathbb{E}_{\zeta_i}[O_i(x; \zeta_i)] = \nabla f_i(x) \quad \text{and} \quad \mathbb{E}_{\zeta_i}[\|O_i(x; \zeta_i) - \nabla f_i(x)\|^2] \leq \sigma^2.$$

### 2.3 Compressor Class

The two classes of widely-used compressors are i) the $\omega$-*unbiased* compressor, described by Assumption 3, *e.g.*, the stochastic quantization operator [4], and ii) the $\delta$-*contractive* compressor, described by Assumption 4, *e.g.*, the rand-$K$ and top-$K$ operators [58, 51].

**Assumption 3 (Unbiased compressor).** *For a (possibly random) compression operator $C : \mathbb{R}^d \to \mathbb{R}^d$, we assume there exists a constant $\omega \geq 0$ such that*

$$\mathbb{E}[C(x)] = x, \quad \mathbb{E}[\|C(x) - x\|^2] \leq \omega\|x\|^2, \quad \forall x \in \mathbb{R}^d,$$

*where the expectation is taken over the randomness of the compression operator $C$.*

**Assumption 4 (Contractive compressor).** *For a (possibly random) compression operator $C : \mathbb{R}^d \to \mathbb{R}^d$, we assume there exists a constant $\delta \in (0, 1]$ such that*

$$\mathbb{E}[\|C(x) - x\|^2] \leq (1 - \delta)\|x\|^2, \quad \forall x \in \mathbb{R}^d,$$

*where the expectation is over the randomness of the compression operator $C$.*

Formally, we let the compressor classes $\mathcal{U}_\omega$ and $\mathcal{C}_\delta$ denote the set of all $\omega$-unbiased compressors and $\delta$-contractive compressors satisfying Assumptions 3 and 4, respectively. Note that the identity operator $I$ satisfies $I \in \mathcal{U}_\omega$ for any $\omega \geq 0$ and $I \in \mathcal{C}_\delta$ for any $\delta \in (0,1]$. Generally, an $\omega$-unbiased compressor is not necessarily contractive when $\omega$ is larger than 1. However, since $C \in \mathcal{U}_\omega$ implies $(1+\omega)^{-1}C \in \mathcal{C}_{(1+\omega)^{-1}}$, the scaled unbiased compressor is also contractive though the converse does not hold. Hence, the class of contractive compressors is strictly more general since it contains all unbiased compressors through scaling.

## 2.4 Algorithm Class

We consider a centralized and synchronous algorithm $A$ in which i) workers are allowed to communicate only directly with the central server but not between one another; ii) all iterations are synchronized so that all workers start each of their iterations simultaneously. Each worker $i$ holds a local copy of the model, denoted by $x_i^{(t)}$, at iteration $t$. The output $\hat{x}^{(t)}$ of $A$ after $t$ iterations can be any linear combination of all previous local models, namely,

$$\hat{x}^{(t)} \in \text{span}\left(\{x_i^{(s)} : 0 \leq s \leq t,\, 1 \leq i \leq n\}\right).$$

We further require algorithms $A$ to satisfy the so-called "zero-respecting" property, which appears in [16, 17, 43]. Intuitively, this property implies that the number of non-zero entries of the local model of a worker can be increased only by conducting local stochastic gradient descent with its own samples or synchronizing with the server. The zero-respecting property holds in all algorithms in Table 1 and most first-order methods based on SGD [46, 35, 31, 79]. In addition to these properties, the algorithm $A$ has to admit communication compression. Specifically, we endow the server with a compressor $C_0$ and each worker $i \in \{1, \cdots, n\}$ with a compressor $C_i$. If $C_i = I$ for some $i \in \{0, \cdots, n\}$, then worker $i$ (or the server if $i = 0$) conducts lossless communication. When $C_i \neq I$ for any $i \in \{0, \cdots, n\}$, algorithm $A$ conducts bidirectional compression. When $C_0 = I$, algorithm $A$ conducts unidirectional compression on messages from workers to server. The formal definition of the algorithm class with bidirectional/unidirectional compression is as follows.

**Definition 1** (**Algorithm class**). *Given compressors $\{C_0, C_1, \ldots, C_n\}$, write $\mathcal{A}^B_{\{C_i\}_{i=0}^n}$ for the set of all centralized, synchronous, zero-respecting algorithms admitting bidirectional compression in which i) compressor $C_i$, $\forall\, 1 \leq i \leq n$, is applied to messages from worker $i$ to the server, and ii) compressor $C_0$ is applied to messages from the server to all workers. When $C_0 = I$, we write $\mathcal{A}^B_{\{C_0=I\} \cup \{C_i\}_{i=1}^n}$, or $\mathcal{A}^U_{\{C_i\}_{i=1}^n}$ for short, for the set of algorithms admitting unidirectional compression. The superscript $B$ or $U$ indicates "bidirectional" or "unidirectional", respectively.*

# 3 Lower Bounds

With all interested classes introduced above, we now define the lower bound measure. Given local loss functions $\{f_i\}_{i=1}^n \subseteq \mathcal{F}_{\Delta,L}$, stochastic gradient oracles $\{O_i\}_{i=1}^n \subseteq \mathcal{O}_{\sigma^2}$ (with $O_i$ for worker $i$), compressors $\{\mathcal{C}_i\}_{i=0}^n \subseteq \mathcal{C}$ ($\mathcal{C}$ can be $\mathcal{U}_\omega$ or $\mathcal{C}_\delta$), and an algorithm $A \in \mathcal{A}$ to solve problem (1) ($\mathcal{A}$ can be $\mathcal{A}^B_{\{C_i\}_{i=0}^n}$ or $\mathcal{A}^U_{\{C_i\}_{i=1}^n}$), we let $\hat{x}_{A, \{f_i\}_{i=1}^n, \{O_i\}_{i=1}^n, \{C_i\}_{i=0}^n, T}$ denote the output of algorithm $A$ using no more than $T$ gradient queries and rounds of communication by each worker node. We define the minimax measure

$$\inf_{A \in \mathcal{A}}\ \sup_{\{C_i\}_{i=0}^n \subseteq \mathcal{C}}\ \sup_{\{O_i\}_{i=1}^n \subseteq \mathcal{O}_{\sigma^2}}\ \sup_{\{f_i\}_{i=1}^n \subseteq \mathcal{F}_{\Delta,L}}\ \mathbb{E}[\|\nabla f(\hat{x}_{A, \{f_i\}_{i=1}^n, \{O_i\}_{i=1}^n, \{C_i\}_{i=0}^n, T})\|^2]. \qquad (2)$$

In (2), we do not require the compressors $\{C_i\}_{i=0}^n$ to be distinct or independent. When $\mathcal{C}$ is $\mathcal{U}_\omega$ or $\mathcal{C}_\delta$, we allow the compressor parameter $\omega$ or $\delta$ to be accessible by algorithm $A$.

## 3.1 Unidirectional Unbiased Compression

Our first result is for algorithms that admit unidirectional compression and $\omega$-unbiased compressors.

**Theorem 1** (**Unidirectional unbiased compression**). *For every $\Delta$, $L > 0$, $n \geq 2$, $\omega \geq 0$, $\sigma > 0$, $T \geq (1+\omega)^2$, there exists a set of local loss functions $\{f_i\}_{i=1}^n \subseteq \mathcal{F}_{\Delta,L}$, stochastic gradient oracles*

$\{O_i\}_{i=1}^n \subseteq \mathcal{O}_{\sigma^2}$, $\omega$-unbiased compressors $\{C_i\}_{i=0}^n \subseteq \mathcal{U}_\omega$ with $C_0 = I$, such that for any algorithm $A \in \mathcal{A}_{\{C_i\}_{i=1}^n}^U$ starting from a given constant $x^{(0)}$, it holds that

$$\mathbb{E}[\|\nabla f(\hat{x}_{A,\{f_i\}_{i=1}^n,\{O_i\}_{i=1}^n,\{C_i\}_{i=0}^n,T})\|^2] = \Omega\left(\left(\frac{\Delta L\sigma^2}{nT}\right)^{\frac{1}{2}} + \frac{(1+\omega)\Delta L}{T}\right). \quad (3)$$

**Consistency with previous works.** The bound in (3) is consistent with best-known lower bounds in different settings. When $\omega = 0$, our result reduces to the tight bounds for distributed training without compression [7]. When $n = 1$ and $\omega = 0$, our result reduces to the lower bound established in [7] under the single-node non-convex stochastic setting. When $n = 1, \omega = 0$ and $\sigma^2 = 0$, our result recovers the tight bound for deterministic non-convex optimization [16].

**Linear-speedup.** When $T$ is sufficiently large, the first term $1/\sqrt{nT}$ dominates the lower bound (3). If an algorithm achieves an $O(1/\sqrt{nT})$ rate, it will require $T = O(1/(n\epsilon^2))$ gradient queries to reach a desired accuracy $\epsilon$, which is inversely proportional to $n$. Therefore, an algorithm achieves linear-speedup at $T$th iteration if, for this $T$, the term involving $nT$ is dominating the rate.

**Transient complexity.** Due to the compression-incurred overhead in convergence rate, a distributed stochastic algorithm with communication compression has to experience a transient stage to achieve its linear-speedup stage. Transient complexity are referred to the number of gradient queries (or communication rounds) when $T$ is relatively small so non-$nT$ terms still dominate the rate. The smaller the transient complexity is, the less gradient queries or communication rounds the algorithm requires to achieve the linear-speedup stage. For example, if an algorithm can achieve the lower bound established in (3), it requires $(\frac{\Delta L\sigma^2}{nT})^{\frac{1}{2}} \geq \frac{(1+\omega)\Delta L}{T}$, i.e., $T = \Omega(n(1+\omega)^2)$ transient gradient queries (or communication rounds) to achieve linear-speedup, which is proportional to the compression-related terms $(1+\omega)^2$. Transient complexity is an important metric to evaluate how sensitive the algorithm is to compression errors. It was widely used in decentralized learning [50, 72] to gauge how the network topology can influence the convergence rate.

### 3.2 Bidirectional Unbiased Compression

Theorem 1 applies to unidirectional compression where $C_0 = I$. We next consider the bidirectional compression with $C_0 \in \mathcal{U}_\omega$. Since $\{C_i\}_{i=1}^n \subseteq \mathcal{U}_\omega$ with $C_0 = I$ is a special case of $\{C_i\}_{i=0}^n \subseteq \mathcal{U}_\omega$, the lower bound for algorithms that admit bidirectional compression is greater than or equal to that with unidirectional compression due to definition in (2).

**Corollary 1** (**Bidirectional unbiased compression**). *Under the same setting as in Theorem 1, there exists a set of local objectives $\{f_i\}_{i=1}^n \subseteq \mathcal{F}_{\Delta,L}$, stochastic gradient oracles $\{O_i\}_{i=1}^n \subseteq \mathcal{O}_{\sigma^2}$, $\omega$-unbiased compressors $\{C_i\}_{i=0}^n \subseteq \mathcal{U}_\omega$ such that for any algorithm $A \in \mathcal{A}_{\{C_i\}_{i=0}^n}^B$ starting from $x^{(0)}$, the lower bound in (3) is also valid.*

Theorem 1 and Corollary 1 indicate that distributed learning with both unidirectional and bidirectional communication compression share the same lower bound. It is intuitive since unidirectional compression is just a special case of bidirectional compression by letting $C_0 = I$.

### 3.3 Unidirectional Contractive Compression

To obtain the lower bounds for communication compression with contractive compressors, we need the following lemma [54, Lemma 1].

**Lemma 1** (**Compressor relation**). *It holds that $\delta\mathcal{U}_{\delta^{-1}-1} \triangleq \{\delta C : C \in \mathcal{U}_{\delta^{-1}-1}\} \subseteq \mathcal{C}_\delta$.*

The above Lemma reveals that any $(\delta^{-1} - 1)$-unbiased compressor is $\delta$-contractive when scaled by $\delta$. Therefore, if an algorithm $A$ admits all $\delta$-contractive compressors, it will also admits all compressors in $\delta\mathcal{U}_{\delta^{-1}-1}$ due to Lemma 1. This relation, together with Theorem 1, helps us achieve the following lower bound with respect to $\delta$-contractive compressors.

**Theorem 2** (**Unidirectional contractive compression**). *For every $\Delta, L > 0$, $n \geq 2$, $0 < \delta \leq 1$, $\sigma > 0$, $T \geq \delta^{-2}$, there exists a set of loss objectives $\{f_i\}_{i=1}^n \subseteq \mathcal{F}_{\Delta,L}$, a set of stochastic gradient oracles $\{O_i\}_{i=1}^n \subseteq \mathcal{O}_{\sigma^2}$, a set of $\delta$-contractive compressors $\{C_i\}_{i=1}^n \subseteq \mathcal{C}_\delta$ with $C_0 = I$, such that*

---

**Algorithm 1** Fast Compressed Communication: $v^{(k,R)} = \text{FCC}(v^{(k,\star)}, C, R, \text{target receiver(s)})$

---

**Input:** The original vector $v^{(k,\star)}$ to communicate at iteration $k$; a compressor $C$
    rounds $R$; initial vector $v^{(k,0)} = 0$; target receiver(s);
**for** $r = 0, \cdots, R-1$ **do**
    Compress $v^{(k,\star)} - v^{(k,r)}$ into $c^{(k,r)} = C(v^{(k,\star)} - v^{(k,r)})$
    Send $c^{(k,r)}$ to the target receiver(s)
    Update $v^{(k,r+1)} = v^{(k,r)} + c^{(k,r)}$
**end for**                    ▷ The set $\{c^{(k,r)}\}_{r=0}^{R-1}$ will be sent to the receiver during the for-loop
**return** Variable $v^{(k,R)}$.          ▷ It holds that $v^{(k,R)} = \sum_{r=0}^{R-1} c^{(k,r)}$

---

*for any algorithm $A \in \mathcal{A}^U_{\{C_i\}_{i=0}^n}$ starting from $x^{(0)}$, it holds that*

$$\mathbb{E}[\|\nabla f(\hat{x}_{A, \{f_i\}_{i=1}^n, \{O_i\}_{i=1}^n, \{C_i\}_{i=0}^n, T})\|^2] = \Omega\left(\left(\frac{\Delta L \sigma^2}{nT}\right)^{\frac{1}{2}} + \frac{\Delta L}{\delta T}\right). \tag{4}$$

**Transient complexity.** With the discussion on transient complexity in Section 3.1, it is easy to derive the transient iteration complexity as $\Omega(n/\delta^2)$ for the lower bound with $\delta$-contractive compressor.

### 3.4 Bidirectional Contractive Compression.

Noting that $\{C_i\}_{i=1}^n \subseteq \mathcal{C}_\delta$ with $C_0 = I$ is a special case of $\{C_i\}_{i=0}^n \subseteq \mathcal{C}_\delta$, we can also establish the lower bound for algorithms that admit bidirectional compression and contractive compressions.

**Corollary 2** (**Bidirectional contractive compression**). *Under the same settings as in Theorem 2, there exists a set of loss objectives $\{f_i\}_{i=1}^n \subseteq \mathcal{F}_{\Delta, L}$, a set of stochastic gradient oracles $\{O_i\}_{i=1}^n \subseteq \mathcal{O}_{\sigma^2}$, a set of $\delta$-contractive compressors $\{C_i\}_{i=1}^n \subseteq \mathcal{C}_\delta$, such that for any algorithm $A \in \mathcal{A}^B_{\{C_i\}_{i=0}^n}$ starting form $x^{(0)}$, the lower bound (4) is also valid.*

## 4 NEOLITHIC: A nearly optimal algorithm

Comparing the best-known upper bounds listed in Table 1 with the established lower bounds in (3) and (4), we find existing algorithms may not be optimal. There exists a clear gap between their convergence rates and our established lower bounds. In this section, we propose NEOLITHIC to fill in this gap. Its rate will match the lower bounds established in (3) and (4) up to logarithm factors. NEOLITHIC can work with both unidirectional and bidirectional compression, and it is compatible with both unbiased and contractive compressors. NEOLITHIC will be discussed in detail with bidirectional contractive compression in this section. It is easy to be adapted to other settings by simply removing the server-to-worker compression or utilizing the scaled unbiased compressor.

### 4.1 Fast Compressed Communication

NEOLITHIC is built on a compression communication protocol listed in Algorithm 1, which we call fast compressed communication (FCC). Given an input vector $v^{(k,\star)}$ to communicate in the $k$-th iteration, FCC will first initialize $v^{(k,0)} = 0$ and then recursively compresses the residual with $c^{(k,r)} \triangleq C(v^{(k,\star)} - v^{(k,r)})$ and sends it to the receiver for $R$ consecutive rounds, see the main recursion in Algorithm 1. When FCC operation finishes, the sender will transmit a set of compressed variables $\{c^{(k,r)}\}_{r=0}^{R-1}$ to the receiver, and return $v^{(k,R)} = \sum_{r=0}^{R-1} c^{(k,r)}$ to itself. Quantity $v^{(k,R)}$ can be regarded as a compressed vector of the original input $v^{(k,\star)}$ after the FCC protocol.

The FCC protocol can be conducted by the server (for which all variables in FCC are without any subscripts, *e.g.*, $c^{(k,r)}$ and $v^{(k,r)}$) or by any worker $i$ (for which all variables are with a subscript $i$, *e.g.*, $c_i^{(k,r)}$ and $v_i^{(k,r)}$). When $R = 1$, FCC reduces to the standard compression utilized in existing literature [62, 58, 24, 68, 32]. While FCC requires $R$ rounds of communication per iteration, the following lemma establishes that the compression error will be exponentially decreased with the number of communication rounds $R$.

---

**Algorithm 2** NEOLITHIC

---

**Input:** Initialize $x^{(0)}$; learning rate $\gamma$; compression round $R$; $\delta^{(-1)} = \delta_i^{(-1)} = 0, \forall i \in [n]$
**for** $k = 0, 1, \cdots, K$ **do**
    **On all workers in parallel:**
        Query stochastic gradients $\hat{g}_i^{(k)} = \frac{1}{R} \sum_{r=0}^{R-1} O_i(x^{(k)}; \zeta_i^{(k,r)})$    ▷ Gradient accumulation
        Error compensate $\tilde{g}_i^{(k)} = \hat{g}_i^{(k)} + \delta_i^{(k-1)}$
        Update error $\delta_i^{(k)} = \tilde{g}_i^{(k)} - \text{FCC}(\tilde{g}_i^{(k)}, C_i, R, \text{server})$    ▷ Worker sends $\{c_i^{(k,r)}\}$ to server
    **On server:**
        Error compensate $\tilde{g}^{(k)} = \delta^{(k-1)} + \frac{1}{n} \sum_{i=1}^{n} \sum_{r=0}^{R-1} c_i^{(k,r)}$   ▷ $\{c_i^{(k,r)}\}$ received from workers
        Update error $\delta^{(k)} = \tilde{g}^{(k)} - \text{FCC}(\tilde{g}^{(k)}, C_0, R, \text{all workers})$ ▷ Server sends $\{c^{(k,r)}\}$ to workers
    **On all workers in parallel:**
        Update model parameter $x^{(k+1)} = x^{(k)} - \gamma \sum_{r=0}^{R-1} c^{(k,r)}$  ▷ $\{c^{(k,r)}\}$ received from server
**end for**

---

**Lemma 2** (FCC property). *Let $C$ be a $\delta$-contractive compressor and $v^{(k,R)} = \text{FCC}(v^{(k,\star)}, C, R)$. It holds for any $R \geq 1$ and $v^{(k,\star)} \in \mathbb{R}^d$ that (proof is in Appendix B)*

$$\mathbb{E}[\|v^{(k,R)} - v^{(k,\star)}\|^2] \leq (1-\delta)^R \|v^{(k,\star)}\|^2, \quad \forall k = 0, 1, 2, \cdots. \tag{5}$$

When $R = 1$, the above FCC property (5) reduces to Assumption 4 for standard contractive compressors. When $R$ is large, FCC can output $v^{(k,R)}$ endowed with very small compression errors. The FCC protocol is also closely related to EF21 compression strategy [52][2]. However, the major novelty of FCC is to utilize multiple such compression rounds to help develop algorithms that can nearly match the established lower bounds.

## 4.2 NEOLITHIC Algorithm

NEOLITHIC is described in Algorithm 2. The FCC protocol in NEOLITHIC communicates $R$ rounds per iteration. To balance the gradient queries and communication rounds, NEOLITHIC will query $R$ stochastic gradients per iteration, see the gradient accumulation step in Algorithm 2. Compared to other algorithms listed in Table 1, the proposed NEOLITHIC takes $R$ times more gradient queries and communication rounds than them per iteration. Given the same budgets to query stochastic gradients and conduct communications as the other standard compression algorithms, we shall consider $K = T/R$ iterations in NEOLITHIC for fair comparison.

We introduce the following assumption to establish the convergence rates of NEOLITHIC.

**Assumption 5** (**Gradient dissimilarity**). *There exists some $b^2 \geq 0$ such that*

$$\frac{1}{n} \sum_{i=1}^{n} \|\nabla f_i(x) - \nabla f(x)\|^2 \leq b^2, \quad \forall x \in \mathbb{R}^d.$$

When local distributions $D_i$ are equivalent across all nodes $i$, we have $f_i(x) = f(x)$ and the above assumption will always hold. We first establish the convergence rate of NEOLITHIC with bidirectional and contractive compressors.

**Theorem 3** (**NEOLITHIC with contractive compressors**). *Given constants $n \geq 1$, $\delta \in (0, 1]$ and Assumption 5, and let $x^{(k)}$ be generated by Algorithm 2. If $R = \lceil \max\{\ln(\delta T \max\{b^2, \sigma^2 \delta\}/\Delta L), \ln(8)\}/\delta \rceil$ and learning rate is set as in Appendix B, it holds for any $K \geq 0$ and compressors $\{C_i\}_{i=0}^{n} \subseteq \mathcal{C}_\delta$ that*

$$\frac{1}{K+1} \sum_{k=0}^{K} \mathbb{E}[\|\nabla f(x^{(k)})\|^2] = \tilde{O}\left( \left(\frac{\Delta L \sigma^2}{nT}\right)^{\frac{1}{2}} + \frac{\Delta L}{\delta T} \right),$$

*where $T = KR$ is the total number of gradient query/communication rounds on each worker and notation $\tilde{O}(\cdot)$ hides logarithmic factors. The above rate implies a transient complexity of $\tilde{O}(n\delta^{-2})$.*

---

[2]If the roles of $v$ and $v^\star$ in Eq. (8) of [52] are switched, we can get the single round of FCC recursions.

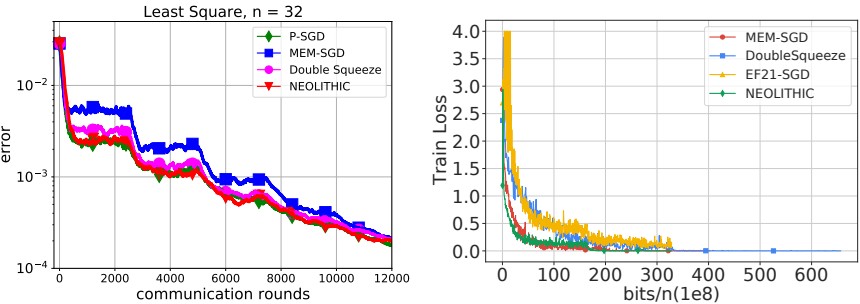

Figure 1: Left: Convergence on the synthetic least square problem in terms of $\mathbb{E}[\|\nabla f(x)\|^2]$ versus communication rounds. Right: Convergence on the CIFAR-10 in terms of training loss versus communication cost.

When the compressor $C$ is replaced with $\omega$-unbiased compressors, we utilize the fact that $(1+\omega)^{-1}C$ is $(1+\omega)^{-1}$-contractive to derive that

**Corollary 3** (**NEOLITHIC with unbiased compressors**). *Under the same assumptions as in Theorem 3, it holds for any $K \geq 0$ and compressors $\{C_i\}_{i=0}^n \subseteq \mathcal{U}_\omega$ that*

$$\frac{1}{K+1}\sum_{k=0}^{K}\mathbb{E}[\|\nabla f(x^{(k)})\|^2] = \tilde{O}\left(\left(\frac{\Delta L \sigma^2}{nT}\right)^{\frac{1}{2}} + \frac{(1+\omega)\Delta L}{T}\right).$$

*This further leads to a transient complexity of $\tilde{O}\left(n(1+\omega)^2\right)$.*

**Remark 1.** *The convergence rates established in Theorem 3 and Corollary 3 are also valid for unidirectional compression when $C_0 = I$. They can match the lower bounds established in Section 3 up to logarithm factors. Moreover, these rates are faster than other algorithms listed in Table 1.*

**Remark 2.** *The results in Theorem 3 and Corollary 3 also imply that NEOLITHIC with bidirectional compression can perform as fast as its counterpart with unidirectional compression. In other words, imposing bidirectional compression can save communications in NEOLITHIC without hurting convergence rates. Before our work, it is established in literature [58, 55, 62, 70, 14] that bidirectional compression leads to slower convergence than unidirectional compression for non-convex problems.*

**Remark 3.** *We remark that Assumption 5 is not required to obtain the lower bounds in Section 3. It is not known whether the lower bounds established in Section 3 can be achieved by NEOLITHIC when Assumption 5 does not hold. However, it is worth noting that Assumption 5 is milder than those made in most works [82, 34, 58, 68, 62, 11] such as bounded gradients. To our best knowledge, only EF21-SGD [24] is guaranteed to converge without Assumption 5, which, however, leads to a fairly loose convergence rate (see Table 1) that cannot show linear-speedup $O(1/\sqrt{nT})$.*

**Remark 4.** *In the deterministic scenario, i.e., $\sigma^2 = 0$, NEOLITHIC produces rate $\tilde{O}((1+\omega)\Delta L/T)$. There exist literature (e.g., [26, 63]) that can achieve rate $\tilde{O}((1 + \omega/n)\Delta L/T)$ which outperforms NEOLITHIC especially when $\omega$ and $n$ are sufficiently large. However, this improvement is built upon an additional assumption that all local compressors $\{C_i\}_{i=0}^n$ are independent of each other and cannot share the same randomness. On the contrary, NEOLITHIC does not impose such an assumption and can be applied to both dependent and independent compressors.*

## 5   Experiments

This section empirically investigates the performance of different compression algorithms with both synthetic simulation and deep learning tasks. We compare NEOLITHIC with PSGD and its variants with communication compression: MEM-SGD, Double-Squeeze, and EF21-SGD. Implementation details and more experiments are provided in Appendix C.

**Linear regression.** We solve a synthetic least-square problem with all aforementioned algorithms. We set $d = 30$, $n = 32$, $R = 4$ (for NEOLITHIC) and utilize the rand-1 compressor. It is observed in Figure 1 (left) that NEOLITHIC beats MEM-SGD and Double-Squeeze with performance close to P-SGD. It implies that algorithms with bidirectional compression can converge as fast as the counterparts with unidirectional compression, which is consistent with results in Remark 1.

Table 2: Accuracy comparison with different algorithms on CIFAR-10 (8 workers, 5% compression ratio).

| METHODS | RESNET18 | RESNET20 |
|---------|----------|----------|
| PSGD | $93.99 \pm 0.52$ | $91.62 \pm 0.13$ |
| MEM-SGD | $94.35 \pm 0.01$ | $91.27 \pm 0.08$ |
| DOUBLE-SQUEEZE | $94.11 \pm 0.14$ | $90.73 \pm 0.02$ |
| EF-21 | $87.37 \pm 0.49$ | $65.82 \pm 4.86$ |
| NEOLITHIC | $\mathbf{94.63 \pm 0.09}$ | $\mathbf{91.43 \pm 0.10}$ |

Table 3: Accuracy comparison with different algorithms on CIFAR-10 (8 workers, 1% compression ratio).

| METHODS | RESNET18 | RESNET20 |
|---------|----------|----------|
| MEM-SGD | $93.99 \pm 0.11$ | $89.68 \pm 0.17$ |
| DOUBLE-SQUEEZE | $93.54 \pm 0.17$ | $89.35 \pm 0.04$ |
| EF-21 | $67.78 \pm 2.14$ | $56.0 \pm 2.257$ |
| NEOLITHIC | $\mathbf{94.155 \pm 0.10}$ | $\mathbf{89.82 \pm 0.37}$ |

**Image classification.** We investigate the performance on two common ResNet models [27] with CIFAR-10 [37] dataset. We train total 300 epochs and set the batch size to 128 on every worker. All experiments were repeated three times with different seeds. For NEOLITHIC, we set $R = 2$. We utilize top-k compressor [62] with different compression ratios. As shown in Figure 1 (right) and Table 2, NEOLITHIC consistently outperforms other compression methods and reach the similar performance to PSGD.

**The effect of compression ratio.** We also investigate the influence of different compression ratios. Table 3 used a compression ratio of 1%, which indicates a more harsh setting for compression methods. NEOLITHIC still outperforms other compression methods.

## 6 Conclusion

This paper provides lower bounds for distributed algorithms with communication compression, whether the compression is unidirectional or bidirectional and unbiased or contractive. An algorithm called NEOLITHIC is introduced to match the lower bounds under the assumption of bounded gradient dissimilarity. Future directions include developing optimal algorithms without the assumption, as well as discovering additional compression properties that might produce a better lower bound.

## Acknowledgements

The authors are grateful to Professor Peter Richtarik from KAUST for the helpful discussions in the relation between FCC and EF21 and various other useful suggestions.

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
