# A  Lower Bounds

In this section, we provide the proofs for Theorem 1 and 2.

## A.1  Proof of Theorem 1

Without loss of generality, we assume algorithms to start from $x^{(0)} = 0$. We denote the $j$-th coordinate of a vector $x \in \mathbb{R}^d$ by $[x]_j$ for $j = 1, \ldots, d$, and let $\mathrm{prog}(x)$ be

$$\mathrm{prog}(x) := \begin{cases} 0 & \text{if } x = 0; \\ \max_{1 \le j \le d} \{j : [x]_j \ne 0\} & \text{otherwise.} \end{cases}$$

Similarly, for a set of points $\mathcal{X} = \{x_1, x_2, \ldots\}$, we define $\mathrm{prog}(\mathcal{X}) := \max_{x \in \mathcal{X}} \mathrm{prog}(x)$. As described in [16, 17], a zero chain function $f$ satisfies

$$\mathrm{prog}(\nabla f(x)) \le \mathrm{prog}(x) + 1, \quad \forall x \in \mathbb{R}^d,$$

which implies that, starting from $x = 0$, a single gradient evaluation can only make at most one more coordinate for the model parameter $x$ be non-zero.

We prove the two terms of the lower bound in Theorem 1 separately by constructing two hard-to-optimize examples. The construction of the example, for each term in (3), can be conducted in three steps: 1) constructing local functions $\{f_i\}_{i=1}^n$ by following the zero-chain function model proposed by [16]; 2) constructing compressors $\{C_i\}_{i=1}^n \in \mathcal{U}_\omega$ and oracles $\{O_i\}_{i=1}^n \subseteq \mathcal{O}_{\sigma^2}$ that hamper algorithms to increase the non-zero coordinates of the model parameters; 3) show a limitation in terms of the non-zero coordinates of model parameters, for algorithms obeying the desired protocol with $T$ gradient queries and compressed communications on each worker, and translate the limitation into the lower bound of convergence rate.

We first state some key zero-chain functions that will be used to facilitate the analysis.

**Lemma 3** (Lemma 2 of [7]). *Let $[x]_j$ denote the $j$-th coordinate of a vector $x \in \mathbb{R}^d$, and define function*

$$h(x) := -\psi(1)\phi([x]_1) + \sum_{j=1}^{d-1} \Big( \psi(-[x]_j)\phi(-[x]_{j+1}) - \psi([x]_j)\phi([x]_{j+1}) \Big)$$

*where for $\forall z \in \mathbb{R}$,*

$$\psi(z) = \begin{cases} 0 & z \le 1/2; \\ \exp\left(1 - \frac{1}{(2z-1)^2}\right) & z > 1/2, \end{cases} \quad \phi(z) = \sqrt{e} \int_{-\infty}^{z} e^{\frac{1}{2}t^2} \mathrm{d}t.$$

*Then $h$ satisfy the following properties:*

1. *$h(x) - \inf_x h(x) \le \delta_0 d$, $\forall x \in \mathbb{R}^d$ with $\delta_0 = 12$.*

2. *$h$ is $\ell_0$-smooth with $\ell_0 = 152$.*

3. *$\|\nabla h(x)\|_\infty \le g_\infty$, $\forall x \in \mathbb{R}^d$ with $g_\infty = 23$.*

4. *$\|\nabla h(x)\|_\infty \ge 1$ for any $x \in \mathbb{R}^d$ with $[x]_d = 0$.*

**Lemma 4.** *Let functions*

$$h_1(x) := -2\psi(1)\phi([x]_1) + 2 \sum_{j \text{ even}, 0 < j < d} \Big( \psi(-[x]_j)\phi(-[x]_{j+1}) - \psi([x]_j)\phi([x]_{j+1}) \Big)$$

*and*

$$h_2(x) := 2 \sum_{j \text{ odd}, 0 < j < d} \Big( \psi(-[x]_j)\phi(-[x]_{j+1}) - \psi([x]_j)\phi([x]_{j+1}) \Big).$$

*Then $h_1$ and $h_2$ satisfy the following properties:*

1. *$\frac{1}{2}(h_1 + h_2) = h$, where $h$ is defined in Lemma 3.*

2. *For any $x \in \mathbb{R}^d$, if $\mathrm{prog}(x)$ is odd, then $\mathrm{prog}(\nabla h_1(x)) \le \mathrm{prog}(x)$; if $\mathrm{prog}(x)$ is even, then $\mathrm{prog}(\nabla h_2(x)) \le \mathrm{prog}(x)$.*

3. *$h_1$ and $h_2$ are also $\ell_0$-smooth with $\ell_0 = 152$.*

*Proof of Lemma 4.* The first property follows the definition of $h_1$, $h_2$, and $h$. The second property follows Lemma 1 of [16] that $\psi^{(m)}(0) = 0$ for any $m \in \mathbb{N}$. Now we prove the third property. Note that the Hessian of $h_k$ for $k = 1, 2$ is tridiagonal and symmetric. Consequently, by the Schur test, we have for any $x \in \mathbb{R}^d$ and $k = 1, 2$

$$\|\nabla^2 h_k(x)\|_2 \le \sqrt{\|\nabla^2 h_k(x)\|_1 \|\nabla^2 h_k(x)\|_\infty} = \|\nabla^2 h_k(x)\|_1. \tag{6}$$

Furthermore, it is easy to verify that

$$\|\nabla^2 h_k(x)\|_1$$
$$\le 2 \max\left\{ \sup_{z \in \mathbb{R}} |\psi''(z)| \sup_{z \in \mathbb{R}} |\phi(z)|, \sup_{z \in \mathbb{R}} |\psi(z)| \sup_{z \in \mathbb{R}} |\phi''(z)| \right\} + 2 \sup_{z \in \mathbb{R}} |\psi'(z)| \sup_{z \in \mathbb{R}} |\phi'(z)| \le 152, \tag{7}$$

where the last inequality follows Observation 2 of [7] that

$$0 \le \psi \le e, \ \ 0 \le \psi' \le \sqrt{54/e}, \ \ |\psi''| \le 32.5, \ \ 0 \le \phi \le \sqrt{2\pi e}, \ \ 0 \le \phi' \le \sqrt{e} \ \text{ and } |\phi''| \le 1.$$

Combining (7) with (6), we know $h_k$ is $\ell_0$-smooth for $k = 1, 2$. $\qquad\square$

Given Lemmas 3 and 4, we now construct two examples that lead to the two terms in lower bound (3), respectively.

**Example 1.**    The proof of the first term $\Omega((\frac{\Delta L \sigma^2}{nT})^{\frac{1}{2}})$ essentially follows the first example in proving Theorem 1 of [43]. We provide the proof for the sake of being self-contained.

(Step 1.) Let $f_i = L\lambda^2 h(x/\lambda)/\ell_0, \forall i = 1, \dots, n$ be homogeneous and hence $f = L\lambda^2 h(x/\lambda)/\ell_0$ where $h$ is defined in Lemma 3 and $\lambda > 0$ is to be specified. Since $\nabla^2 f_i = L\nabla^2 h/\ell_0$ and $h$ is $\ell_0$-smooth (Lemma 3), we know $f_i$ is $L$-smooth for any $\lambda > 0$. By Lemma 3, we have

$$f(0) - \inf_x f(x) = \frac{L\lambda^2}{\ell_0^2}(h(0) - \inf_x h(x)) \le \frac{L\lambda^2 \delta_0 d}{\ell_0}.$$

Therefore, to ensure $f_i \in \mathcal{F}_{\Delta, L}$, it suffices to let

$$\frac{L\lambda^2 \delta_0 d}{\ell_0} \le \Delta, \quad \text{i.e.,} \quad d\lambda^2 \le \frac{\ell_0 \Delta}{L\delta_0}. \tag{8}$$

(Step 2.) We consider all compressors $\{C_i\}_{i=1}^n$ are identity, meaning that there is no compression error in the optimization procedure. Then we naturally have $\{C_i\}_{i=1}^n \subseteq \mathcal{U}^\omega$ for any $\omega \ge 0$. We construct the stochastic gradient oracle $O_i$ on worker $i, \forall i = 1, \dots, n$ as the following:

$$[O_i(x; Z)]_j = [\nabla f_i(x)]_j \left(1 + \mathbb{1}\{j > \mathrm{prog}(x)\}\left(\frac{Z}{p} - 1\right)\right), \forall x \in \mathbb{R}^d, j = 1, \dots, d$$

with random variable $Z \sim \mathrm{Bernoulii}(p)$ independent of $x$ and $f_i$, and $p \in (0, 1)$ to be specified. The oracle $O_i$ has probability to make the entry $[\nabla f_i(x)]_{\mathrm{prog}(x)+1}$ zero. Therefore algorithms are hampered by the gradient oracle to achieve more non-zero coordinates for the model parameters. It is easy to see $O_i$ is an unbiased stochastic gradient oracle. Moreover, since $f_i$ is zero-chain, we have $\mathrm{prog}(O_i(x; Z)) \le \mathrm{prog}(\nabla f_i(x)) \le \mathrm{prog}(x) + 1$ and hence

$$\mathbb{E}[\|[O_i(x; Z)] - \nabla f_i(x)\|^2] = |[\nabla f_i(x)]_{\mathrm{prog}(x)+1}|^2 \mathbb{E}\left[\left(\frac{Z}{p} - 1\right)^2\right] = |[\nabla f_i(x)]_{\mathrm{prog}(x)+1}|^2 \frac{1-p}{p}$$

$$\le \|\nabla f_i(x)\|_\infty^2 \frac{1-p}{p} \le \frac{L^2\lambda^2(1-p)}{\ell_0^2 p} \|\nabla h(x)\|_\infty^2$$

$$\overset{\text{Lemma 3}}{\le} \frac{L^2\lambda^2(1-p)g_\infty^2}{\ell_0^2 p}.$$

Therefore, to ensure $O_i \in \mathcal{O}_{\sigma^2}$, it suffices to let

$$p = \min\{\frac{L^2\lambda^2 g_\infty^2}{\ell_0^2\sigma^2}, 1\}. \tag{9}$$

(Step 3.) Let $x_i^{(t)}, \forall t = 0,\dots$ and $1 \leq i \leq n$, be the $t$-th query point of worker $i$. Since algorithms satisfy the zero-respecting property, as discussed in [16, 17, 43], within $T$ gradient queries on each worker, algorithms can only return model $\hat{x}$ such that

$$\hat{x} \in \mathrm{span}\left(\left\{x^{(0)}, \nabla f_i(x^{(0)}), \{\{x_i^{(t)}, \nabla f_i(x_i^{(t)}) : 0 \leq t < T\} : 1 \leq i \leq n\}\right\}\right),$$

which implies

$$\mathrm{prog}(\hat{x}) \leq \max_{0 \leq t < T} \max_{1 \leq i \leq n} \mathrm{prog}(x_i^{(t)}) + 1. \tag{10}$$

By Lemma 2 of [43], we have

$$\mathbb{P}(\mathrm{prog}(\hat{x}) \geq d) \leq \mathbb{P}(\max_{0 \leq t < T} \max_{1 \leq i \leq n} \mathrm{prog}(x_i^{(t)}) \geq d - 1) \leq e^{(e-1)npT-d+1}. \tag{11}$$

On the other hand, when $\mathrm{prog}(\hat{x}) < d$, by the fourth point in Lemma 3, it holds that

$$\min_{\hat{x} \in \mathrm{span}\{\{x_i^{(t)} : 1 \leq i \leq n, 0 \leq t < T\}\}} \|\nabla f(\hat{x})\| \geq \min_{[\hat{x}]_d = 0} \|\nabla f(\hat{x})\| = \frac{L\lambda}{\ell_0} \min_{[\hat{x}]_d = 0} \|\nabla h(\hat{x})\| \geq \frac{L\lambda}{\ell_0}. \tag{12}$$

Therefore, by combining (11) and (12), we have

$$\mathbb{E}[\|\nabla f(\hat{x})\|^2] \geq \mathbb{P}(\mathrm{prog}^{(T)} < d)\mathbb{E}[\|\nabla f(\hat{x})\|^2 \mid \mathrm{prog}^{(T)} < d] \geq (1 - e^{(e-1)npT-d+1})\frac{L^2\lambda^2}{\ell_0^2}. \tag{13}$$

Let

$$\lambda = \frac{\ell_0}{L}\left(\frac{\Delta L\sigma^2}{3nT\ell_0\delta_0 g_\infty^2}\right)^{\frac{1}{4}} \quad \text{and} \quad d = \left\lfloor\left(\frac{3L\Delta nTg_\infty^2}{\sigma^2\ell_0\delta_0}\right)^{\frac{1}{2}}\right\rfloor. \tag{14}$$

Then (8) naturally holds and $p = \min\{\frac{g_\infty^2}{\sigma^2}\left(\frac{\Delta L\sigma^2}{3nT\ell_0\delta_0 g_\infty^2}\right)^{\frac{1}{2}}, 1\}$ by (9). Without loss of generality, we assume $d \geq 2$, which is guaranteed when $T = \Omega(\frac{\sigma^2}{nL\Delta})$. Then, using the definition of $p$, we have that

$$(e-1)npT - d + 1 \leq (e-1)nT\frac{g_\infty^2}{\sigma^2}\left(\frac{\Delta L\sigma^2}{3nT\ell_0\delta_0 g_\infty^2}\right)^{\frac{1}{2}} - d + 1$$

$$= \frac{e-1}{3}\left(\frac{3L\Delta nTg_\infty^2}{\sigma^2\ell_0\delta_0}\right)^{\frac{1}{2}} - d + 1 < \frac{e-1}{3}(d+1) - d + 1 \leq 2 - e < 0$$

which, combined with (13), further implies

$$\mathbb{E}[\|\nabla f(\hat{x})\|^2] = \Omega\left(\frac{L^2\lambda^2}{\ell_0^2}\right) = \Omega\left(\left(\frac{\Delta L\sigma^2}{3nT\ell_0\delta_0 g_\infty^2}\right)^{\frac{1}{2}}\right) = \Omega\left(\left(\frac{\Delta L\sigma^2}{nT}\right)^{\frac{1}{2}}\right).$$

**Example 2.** Without loss of generality, we assume $n$ is even, otherwise we can consider the lower bound for the case of $n - 1$.

(Step 1.) Similar to the construction of Example 1, we let $f_i = L\lambda^2 h_1(x/\lambda)/\ell_0, \forall 1 \leq i \leq n/2$ and $f_i = L\lambda^2 h_2(x/\lambda)/\ell_0, \forall n/2 < i \leq n$, where $h_1$ and $h_2$ are defined in Lemma 4, and $\lambda > 0$ will be specified later. By the definitions of $h_1$ and $h_2$, we have that $f_i, \forall 1 \leq i \leq n$ is zero-chain and $f(x) = \frac{1}{n}\sum_{i=1}^n f_i(x) = L\lambda^2/\ell_0 h(x/\lambda)$. Since $h_1$ and $h_2$ are also $\ell_0$-smooth, to let $f \in \mathcal{F}_{\Delta,L}$, it suffices to make (8) hold.

In the above construction, we essentially split a zero-chain function, i.e., $h$, into two different components: the even component of the chain, i.e., $h_1$ and the odd component of the chain, i.e., $h_2$. Recall the second property in Lemma 4 that for any $x \in \mathbb{R}^d$, if $\mathrm{prog}(x)$ is odd, then $\mathrm{prog}(\nabla h_1(x)) \leq \mathrm{prog}(x)$; if $\mathrm{prog}(x)$ is even, then $\mathrm{prog}(\nabla h_2(x)) \leq \mathrm{prog}(x)$. Therefore, the workers, starting from

any point with any algorithm $A \in \mathcal{A}^U_{\{C_i\}^n_{i=1}}$, can only earn one more non-zero coordinate if they do not synchronize (communicate) with the server; after that, the number of non-zero coordinates of local models will not increase any more. That is to say, in order to proceed (i.e.,, achieve more non-zero coordinates), the worker must synchronize the gradient information, via the server, between the odd and even components.

(Step 2.) We consider a gradient oracle that can return the lossless full-batch gradient, i.e., $O_i(x) = \nabla f_i(x), \forall x \in \mathbb{R}, 1 \leq i \leq n$. For the construction of $\omega$-unbiased compressors, we consider $\{C_i\}^n_{i=1}$ to be the $\frac{d}{s} \times$rand-$s$ operators with shared randomness and $s = \lceil d/(1+\omega) \rceil$, where the $\frac{d}{s}$-scaling procedure is to ensure unbiasedness. Specifically, during a round of communication, all workers randomly choose $s$ coordinates from the full vector to be communicated, and then transmit the $\frac{d}{s}$-scaled values at chosen coordinates. The chosen coordinate indexes are identical across all workers due to the shared randomness and are sampled uniformly randomly per communication. Since for any $1 \leq k \leq d$, the index $k$ has probability $s/d$ to be chosen, we have for any $x \in \mathbb{R}^d$

$$\mathbb{E}[C_i(x)] = \mathbb{E}\left[\left(\frac{d}{s}x_k \mathbb{1}\{k \text{ is chosen}\}\right)_{1 \leq k \leq d}\right] = \left(\frac{d}{s}x_i \mathbb{P}(k \text{ is chosen})\right)_{1 \leq k \leq d} = x,$$

and

$$\mathbb{E}[\|C_i(x) - x\|^2] = \sum_{k=1}^{d} \mathbb{E}\left[\left(\frac{d}{s}x_k \mathbb{1}\{k \text{ is chosen}\} - x_k\right)^2\right]$$

$$= \sum_{k=1}^{d} x_k^2 \left(\left(\frac{d}{s} - 1\right)^2 \mathbb{P}(k \text{ is chosen}) + \mathbb{P}(k \text{ is not chosen})\right) = \sum_{k=1}^{d} x_k^2 \left(\frac{d}{s} - 1\right) \leq \omega\|x\|^2$$

where the last inequality follows the definition of $s$. Therefore, the above construction ensures $\{C_i\}^n_{i=1} \subseteq \mathcal{U}_\omega$.

For any $t = 0, 1, \ldots$ and $1 \leq i \leq n$, let $v_i^{(t,\star)}$ be the synchronized point that worker $i$ wants to send to the server at the $t$-th communication. Due to communication compression, the server can only receive a compressed version $C_i(v_i^{(t,\star)})$ instead, which we denote by $v_i^{(t)} \triangleq C_i(v_i^{(t,\star)})$. It is easy to see that

$$\text{prog}(v_i^{(t)}) \leq \text{prog}(v_i^{(t,\star)}), \quad \forall v_i^{(t,\star)} \in \mathbb{R}^d. \tag{15}$$

Furthermore, since the compressor $C_i$ only takes $s$ coordinates, $v_i^{(t)}$ has probability $1 - s/d \approx \frac{\omega}{1+\omega}$ in making the coordinate with index $\text{prog}(v_i^{(t,\star)})$ zero, which implies $\mathbb{P}(\text{prog}(v^{(t)}) < \text{prog}(v^{(t,\star)})) \approx \frac{\omega}{1+\omega}$. This is to say, each worker $i$ has only probability $\approx (1+\omega)^{-1}$ to transmit its last non-zero entry. Therefore, algorithms are hampered by the compressors to achieve more non-zero coordinates for model parameters in synchronizing the gradient information from all workers.

(Step 3.) Since we only consider algorithms satisfying the zero-respecting property, as discussed in [16, 17, 43], worker $i$, within $T$ rounds of communication, can only return model $\hat{x}$ such that

$$\hat{x} \in \text{span}\left(\left\{x^{(0)}, \nabla f_i(x^{(0)}), \{\{v_i^{(t)}, \nabla f_i(v_i^{(t)}) : 0 \leq t < T\} : 1 \leq i \leq n\}\right\}\right). \tag{16}$$

Since $f_i$s are zero-chain functions, $\hat{x}_i$ satisfies

$$\text{prog}(\hat{x}) \leq \text{prog}\left(\left\{x^{(0)}, \nabla f_i(x^{(0)}), \{\{v_i^{(t)}, \nabla f_i(v_i^{(t)}) : 0 \leq t < T\} : 1 \leq i \leq n\}\right\}\right)$$

$$\leq \max_{0 \leq t < T} \max_{1 \leq i \leq n} \text{prog}(v_i^{(t)}) + 1. \tag{17}$$

By Lemma 5, we have

$$\mathbb{P}\left(\max_{0 \leq t < T} \max_{1 \leq i \leq n} \text{prog}(v_i^{(t)}) \geq d - 1\right) \leq e^{(e-1)T\lceil d/(1+\omega) \rceil/d + 1 - d}. \tag{18}$$

Combining (18) with (17) and (12), we have that

$$\mathbb{E}[\|\nabla f(\hat{x})\|^2] \geq (1 - e^{(e-1)T\lceil d/(1+\omega) \rceil/d + 1 - d})\frac{L^2\lambda^2}{\ell_0^2}. \tag{19}$$

Let

$$\lambda = \frac{\ell_0}{L}\sqrt{\frac{(1+\omega)\Delta L}{5T\ell_0\delta_0}} \quad \text{and} \quad d = \lfloor 5T/(1+\omega)\rfloor. \tag{20}$$

Then (8) naturally holds. Since $T$ is assumed to be no less than $(1+\omega)^2$, we have $d = \lfloor 5T/(1+\omega)\rfloor \geq 5T/(1+\omega) - 1 \geq 4T/(1+\omega) \geq 4(1+\omega) \geq 4$. Then it is easy to verify

$$(e-1)T\left\lceil\frac{d}{1+\omega}\right\rceil/d + 1 - d \leq (e-1)T\left(\frac{d}{1+\omega}+1\right)/d + 1 - d$$

$$=(e-1)\frac{T}{1+\omega} + (e-1)T/d + 1 - d \leq (e-1)\frac{T}{1+\omega} + (e-1)\frac{T}{4(1+\omega)} + 1 - \frac{4T}{1+\omega}$$

$$=((e-1)(1+\frac{1}{4})-4)\frac{T}{1+\omega} + 1 \leq \frac{5e-17}{4} < 0,$$

which, combined with (19), further implies

$$\mathbb{E}[\|\nabla f(\hat{x})\|^2] = \Omega\left(\frac{L^2\lambda^2}{\ell_0^2}\right) = \Omega\left(\frac{\Delta L}{5\delta T\ell_0\delta_0}\right) = \Omega\left(\frac{(1+\omega)\Delta L}{T}\right).$$

**Lemma 5.** *In Example 2 in the proof of Theorem 1, it holds that*

$$\mathbb{P}(\max_{0\leq t<T}\max_{1\leq i\leq n}\mathrm{prog}(v_i^{(t)}) \geq d-1) \leq e^{(e-1)T\lceil d/(1+\omega)\rceil/d-d}$$

*for any $T \geq 0$.*

*Proof.* Note that for any $0 \leq t < T$, at the $(t+1)$-th round of communication, worker $i$ can only transmit vector $v_i^{(t,\star)}$ that is aggregated from local gradient descent and the vectors in the past communication. Therefore, $v_i^{(t,\star)}$ satisfies

$$\mathrm{prog}(v_i^{(t,\star)}) \leq \mathrm{prog}\left(\left\{x^{(0)}, \nabla f_i(x^{(0)}), \{\{v_i^{(s)}, \nabla f_i(v_i^{(s)}) : 0 \leq s < t\} : 1 \leq i \leq n\}\right\}\right)$$

$$\leq \max_{0\leq s<t}\max_{1\leq i\leq n}\mathrm{prog}(v_i^{(s)}) + 1 \triangleq B^{(t)}. \tag{21}$$

We define $B^{(0)} = 1$ additionally. By the definition of $B^{(t)}$ and that $\mathrm{prog}(v^{(t,\star)}) \leq \mathrm{prog}(v_i^{(t,\star)})$, it naturally holds that

$$B^{(t)} \leq B^{(t+1)} = \max_{0\leq s<(t+1)}\max_{1\leq i\leq n}\mathrm{prog}(v_i^{(s)}) + 1 = \max\left\{B^{(t)}, \max_{1\leq i\leq n}\mathrm{prog}(v_i^{(t)})\right\} + 1$$

$$\leq \max\left\{B^{(t)}, \max_{1\leq i\leq n}\mathrm{prog}(v_i^{(t,\star)})\right\} + 1 \overset{(21)}{\leq} B^{(t)} + 1. \tag{22}$$

Therefore, one round of communication can increase $B^{(t)}$ at most by 1.

From (22), we see that $B^{(t+1)} = B^{(t)} + 1$ only when $\max_{1\leq i\leq n}\mathrm{prog}(v_i^{(t)}) = \max_{1\leq i\leq n}\mathrm{prog}(v_i^{(t,\star)})$. Let $k = \max_{1\leq i\leq n}\mathrm{prog}(v_i^{(t,\star)})$. Recall that the compressors constructed in Example 2 are built on the shared randomness, therefore $\max_{1\leq i\leq n}\mathrm{prog}(v_i^{(t)}) = \max_{1\leq i\leq n}\mathrm{prog}(v_i^{(t,\star)}) = k$ is equivalent to that the coordinate index $k$ is chosen during the $(t+1)$-round of communication compression, which is of probability $\frac{s}{d}$. Therefore, we have

$$\mathbb{P}(B^{(t+1)} = B^{(t)} + 1) \leq \mathbb{P}(\max_{1\leq i\leq n}\mathrm{prog}(v_i^{(t)}) = \max_{1\leq i\leq n}\mathrm{prog}(v_i^{(t,\star)}))$$

$$=\mathbb{P}\left(\text{the coordinate index } \max_{1\leq i\leq n}\mathrm{prog}(v_i^{(t,\star)}) \text{ is chosen}\right) = \frac{s}{d}. \tag{23}$$

Let $E^{(t+1)}$ be the event {the coordinate index $\max_{1\leq i\leq n}\mathrm{prog}(v_i^{(t,\star)})$ is chosen during the $(t+1)$-round of compression}. Since the compression is uniformly random, we have

$\mathbb{1}(E^{(1)}), \ldots, \mathbb{1}(E^{(T)}) \overset{\text{i.i.d.}}{\sim} \text{Bernoulli}(\frac{s}{d})$ where $\mathbb{1}(\cdot)$ is the indicator function. By the above argument, we also have $B^{(t+1)} - B^{(t)} \leq \mathbb{1}(E^{(t+1)})$ for any $0 \leq t < T$. Therefore, we have

$$\mathbb{P}(B^{(T)} \geq d) \leq e^{-d}\mathbb{E}[e^{B^{(T)}}] = e^{-d}\mathbb{E}\left[\exp\left(B^{(0)} + \sum_{t=0}^{T-1}(B^{(t+1)} - B^{(t)})\right)\right]$$

$$\leq e^{-(d-1)}\mathbb{E}\left[\exp\left(\sum_{t=0}^{T-1}\mathbb{1}(E^{(t+1)})\right)\right] = e^{-(d-1)}\prod_{t=0}^{T-1}\mathbb{E}\left[\exp\left(\mathbb{1}(E^{(t+1)})\right)\right]$$

$$= e^{-(d-1)}\prod_{t=0}^{T-1}\left(1 + \frac{s}{d}(e-1)\right) \leq e^{-(d-1)}\prod_{t=0}^{T-1}e^{(e-1)s/d} = e^{(e-1)Ts/d-d+1},$$

which directly leads to the conclusion. $\qquad\square$

### A.2 Proof of Theorem 2

Theorem 2 essentially follows the same analysis as in Theorem 1. The only difference is that we shall construct compressors in proving $\Omega(\frac{\Delta L}{\delta T})$ by using rand-$s$ operators with shared randomness and $s = \lceil \delta d \rceil$. There is no scaling procedure in compression. We can easily check that

$$\mathbb{E}[\|C_i(x) - x\|^2] = \sum_{k=1}^{d}\mathbb{E}\left[(x_k\mathbb{1}\{k \text{ is chosen}\} - x_k)^2\right]$$

$$= \sum_{k=1}^{d}x_k^2\mathbb{P}(k \text{ is not chosen}) = \left(1 - \frac{s}{d}\right)\|x\|^2 \leq \delta\|x\|^2.$$

where the last inequality follows the definition of $s$. Therefore, we have $\{C_i\}_{i=1}^{n} \subseteq \mathcal{C}_\delta$. The scaling procedure does not change prog, and thus makes no effect on the argument in terms of non-zero coordinates. By considering $\omega = \delta^{-1} - 1$, i.e., $1 + \omega = \delta$, we can easily adapt the proof of Theorem 1 to reach Theorem 2.

## B Convergence of NEOLITHIC

### B.1 Proof of Lemma 2

Let $v^{(k,r)}$ be the $r$-th intermediate point generated in FCC for any $0 \leq r \leq R$. Since $C$ is $\delta$-contractive, we have

$$\mathbb{E}[\|v^{(k,R)} - v^{(k,\star)}\|^2] = \mathbb{E}[\|v^{(k,R-1)} + C(v^{(k,\star)} - v^{(k,R-1)}) - v^{(k,\star)}\|^2]$$

$$= \mathbb{E}\left[\mathbb{E}[\|C(v^{(k,\star)} - v^{(k,R-1)}) + v^{(k,R-1)} - v^{(k,\star)}\|^2 \mid v^{(k,R-1)}]\right]$$

$$\leq (1-\delta)\mathbb{E}[\|v^{(k,R-1)} - v^{(k,\star)}\|^2]$$

$$\leq (1-\delta)^2\mathbb{E}[\|v^{(k,R-2)} - v^{(k,\star)}\|^2]$$

$$\leq (1-\delta)^R\mathbb{E}[\|v^{(k,0)} - v^{(k,\star)}\|^2] = (1-\delta)^R\|v^{(k,\star)}\|^2.$$

### B.2 Proof of Theorem 3

In this subsection, we provide the convergence proof for NEOLITHIC with bidirectional compression and contractive compressors. We first introduce some notations: $\Omega^{(k)} := \delta^{(k)} + \frac{1}{n}\sum_{i=1}^{n}\delta_i^{(k)}$, $\forall k \geq -1$; $y^{(k)} := x^{(k)} - \gamma\Omega^{(k-1)}, \forall k \geq 0$; $\Psi^{(k)} := \|\delta^{(k)}\|^2 + \frac{3}{n}\sum_{i=1}^{n}\|\delta_i^{(k)}\|^2, \forall k \geq -1$. We will use the following lemmas.

**Lemma 6** (RECURSION FORMULA). *For any $k \geq 0$, it holds that*

$$x^{(k+1)} - x^{(k)} = -\frac{\gamma}{n}\sum_{i=1}^{n}\hat{g}_i^{(k)} - \gamma\Omega^{(k-1)} + \gamma\Omega^{(k)}.$$

*Proof.* It is observed from Algorithm 1 that the vector $v^{(k,R)}$ returned by the FCC operator satisfies

$$v^{(k,R)} = \sum_{r=0}^{R-1} c^{(k,r)} \quad (\text{or } v_i^{(k,R)} = \sum_{r=0}^{R-1} c_i^{(k,r)} \text{ if FCC is utilized in node } i) \tag{24}$$

With (24), the relation between $\tilde{g}^{(k)}$ and $\delta^{(k)}$ (or between $\tilde{g}_i^{(k)}$ and $\delta_i^{(k)}$) in Algorithm 2 satisfies

$$\tilde{g}^{(k)} - \delta^{(k)} = \sum_{r=0}^{R-1} c^{(k,r)} \quad \text{and} \quad \tilde{g}_i^{(k)} - \delta_i^{(k)} = \sum_{r=0}^{R-1} c_i^{(k,r)}, \quad \forall 1 \le i \le n. \tag{25}$$

Therefore, we have for any $k \ge 0$

$$\begin{aligned}
x^{(k+1)} - x^{(k)} &= -\gamma \sum_{r=0}^{R-1} c^{(k,r)} \overset{(25)}{=} -\gamma(\tilde{g}^{(k)} - \delta^{(k)}) \\
&= -\gamma \left( \delta^{(k-1)} + \frac{1}{n} \sum_{i=1}^{n} \sum_{r=0}^{R-1} c_i^{(k,r)} - \delta^{(k)} \right) \tag{26} \\
&\overset{(25)}{=} -\gamma \left( \delta^{(k-1)} + \frac{1}{n} \sum_{i=1}^{n} \left( \tilde{g}_i^{(k)} - \delta_i^{(k)} \right) - \delta^{(k)} \right) \\
&= -\gamma \left( \delta^{(k-1)} + \frac{1}{n} \sum_{i=1}^{n} \left( \hat{g}_i^{(k)} + \delta_i^{(k-1)} - \delta_i^{(k)} \right) - \delta^{(k)} \right) \tag{27} \\
&= -\frac{\gamma}{n} \sum_{i=1}^{n} \hat{g}_i^{(k)} - \gamma \Omega^{(k-1)} + \gamma \Omega^{(k)} \tag{28}
\end{aligned}$$

where (26) and (27) follow the implementation of Algorithm 2, and we use the notation of $\Omega^{(k)}$ in (28). $\qquad\square$

**Lemma 7** (DESCENT LEMMA). *Let the auxiliary sequence be $y^{(k)} := x^{(k)} - \gamma \Omega^{(k-1)}$, $\forall k \ge 0$. Under Assumption 1, if learning rate $0 < \gamma \le \frac{1}{2L}$, it holds that for any $k \ge 0$,*

$$\mathbb{E}[f(y^{(k+1)})] \le \mathbb{E}[f(y^{(k)})] - \frac{\gamma}{4} \mathbb{E}[\|\nabla f(x^{(k)})\|^2] + 2\gamma^3 L^2 \mathbb{E}[\|\Omega^{(k-1)}\|^2] + \frac{\gamma^2 L \sigma^2}{2nR}. \tag{29}$$

*Proof.* By Lemma 6 and the definition of $y^{(k)}$ for $k \ge 0$, we directly have that

$$y^{(k+1)} = y^{(k)} - \frac{\gamma}{n} \sum_{i=1}^{n} \hat{g}_i^{(k)}.$$

Since $f$ is $L$-smooth, we have

$$\begin{aligned}
f(y^{(k+1)}) &\le f(y^{(k)}) + \langle \nabla f(y^{(k)}), y^{(k+1)} - y^{(k)} \rangle + \frac{L}{2} \|y^{(k+1)} - y^{(k)}\|^2 \\
&= f(y^{(k)}) - \gamma \left\langle \nabla f(y^{(k)}), \frac{1}{n} \sum_{i=1}^{n} \hat{g}_i^{(k)} \right\rangle + \frac{\gamma^2 L}{2} \left\| \frac{1}{n} \sum_{i=1}^{n} \hat{g}_i^{(k)} \right\|^2. \tag{30}
\end{aligned}$$

Since $\hat{g}_i^{(k)} = \frac{1}{R} \sum_{r=1}^{R} O_i(x^{(k)}; \zeta_i^{(k,r)})$ is a unbiased estimator of $\nabla f_i(x^{(k)})$, and by Assumption 2, we have

$$\mathbb{E} \left[ \frac{1}{n} \sum_{i=1}^{n} \hat{g}_i^{(k)} \right] = \frac{1}{n} \sum_{i=1}^{n} \nabla f_i(x^{(k)}) = \nabla f(x^{(k)}) \tag{31}$$

and

$$\mathbb{E}\left[\left\|\frac{1}{n}\sum_{i=1}^{n}\hat{g}_i^{(k)}\right\|^2\right] = \|\nabla f(x^{(k)})\|^2 + \mathbb{E}\left[\left\|\frac{1}{n}\sum_{i=1}^{n}\hat{g}_i^{(k)} - \nabla f(x^{(k)})\right\|^2\right]$$

$$= \|\nabla f(x^{(k)})\|^2 + \mathbb{E}\left[\left\|\frac{1}{nR}\sum_{i=1}^{n}\sum_{r=0}^{R-1}\left(O_i(x^{(k)};\zeta_i^{(k,r)}) - \nabla f_i(x^{(k)})\right)\right\|^2\right]$$

$$\leq \|\nabla f(x^{(k)})\|^2 + \frac{\sigma^2}{nR}. \tag{32}$$

Taking global expectation over (30), and plugging (31) and (32) into it, we reach

$$\mathbb{E}[f(y^{(k+1)})] - \mathbb{E}[f(y^{(k)})]$$

$$\leq -\gamma\mathbb{E}[\langle\nabla f(y^{(k)}), \nabla f(x^{(k)})\rangle] + \frac{\gamma^2 L}{2}\mathbb{E}[\|\nabla f(x^{(k)})\|^2] + \frac{\gamma^2 L\sigma^2}{2nR}$$

$$= -\gamma\mathbb{E}[\langle\nabla f(y^{(k)}) - \nabla f(x^{(k)}), \nabla f(x^{(k)})\rangle] - \left(\gamma - \frac{\gamma^2 L}{2}\right)\mathbb{E}[\|\nabla f(x^{(k)})\|^2] + \frac{\gamma^2 L\sigma^2}{2nR}$$

$$\leq 2\gamma\mathbb{E}[\|\nabla f(y^{(k)}) - \nabla f(x^{(k)})\|^2] + \frac{\gamma}{2}\mathbb{E}[\|\nabla f(x^{(k)})\|^2]$$

$$- \left(\gamma - \frac{\gamma^2 L}{2}\right)\mathbb{E}[\|\nabla f(x^{(k)})\|^2] + \frac{\gamma^2 L\sigma^2}{2nR} \tag{33}$$

$$\leq 2\gamma L^2\mathbb{E}[\|y^{(k)} - x^{(k)}\|^2] - \frac{\gamma(1-\gamma L)}{2}\mathbb{E}[\|\nabla f(x^{(k)})\|^2] + \frac{\gamma^2 L\sigma^2}{2nR}, \tag{34}$$

where we use Young's inequality in (33), and (34) holds by Assumption 1. Using $y^{(k)} - x^{(k)} = -\gamma\Omega^{(k-1)}$ and that

$$0 < \gamma \leq \frac{1}{2L} \quad\Longrightarrow\quad \frac{\gamma(1-\gamma L)}{2} \geq \frac{\gamma}{4}$$

in (34), we reach the conclusion in this lemma. $\qquad\square$

**Lemma 8** (VANISHING ERROR). *Assume $R$ is sufficiently large such that $(1-\delta)^R < \frac{1}{4}$ and let $\Psi^{(k)} := \|\delta^{(k)}\|^2 + \frac{3}{n}\sum_{i=1}^{n}\|\delta_i^{(k)}\|^2$ for $k \geq -1$. Under Assumptions 2, 4, 5 It holds that*

$$\mathbb{E}[\Psi^{(k)}] \leq 4(1-\delta)^R\left(\mathbb{E}[\Psi^{(k-1)}] + 5\mathbb{E}[\|\nabla f(x^{(k)})\|^2] + 4\left(b^2 + \frac{\sigma^2}{R}\right)\right).$$

*Proof.* By Lemma 2, it holds that

$$\mathbb{E}[\|\delta^{(k)}\|^2] = \mathbb{E}[\|\tilde{g}^{(k)} - \text{FCC}(\tilde{g}^{(k)}, C, R)\|^2] \leq (1-\delta)^R\mathbb{E}[\|\tilde{g}^{(k)}\|^2].$$

Note that

$$\tilde{g}^{(k)} = \delta^{(k-1)} + \frac{1}{n}\sum_{i=1}^{n}\sum_{r=0}^{R-1}c_i^{(k,r)} = \delta^{(k-1)} + \frac{1}{n}\sum_{i=1}^{n}\left(\tilde{g}_i^{(k)} - \delta_i^{(k)}\right)$$

$$= \delta^{(k-1)} + \frac{1}{n}\sum_{i=1}^{n}\left(\hat{g}_i^{(k)} + \delta_i^{(k-1)} - \delta_i^{(k)}\right)$$

$$= \delta^{(k-1)} + \frac{1}{n}\sum_{i=1}^{n}\hat{g}_i^{(k)} + \frac{1}{n}\sum_{i=1}^{n}\delta_i^{(k-1)} - \frac{1}{n}\sum_{i=1}^{n}\delta_i^{(k)}.$$

Therefore, by using Young's inequality and (32), we have

$$\mathbb{E}[\|\delta^{(k)}\|^2] \leq (1-\delta)^R \mathbb{E}[\|\tilde{g}^{(k)}\|^2]$$

$$\leq (1-\delta)^R \left( 4\mathbb{E}[\|\delta^{(k-1)}\|^2] + 4\mathbb{E}\left[ \left\| \frac{1}{n}\sum_{i=1}^{n} \hat{g}_i^{(k)} \right\|^2 \right] \right.$$

$$+ \frac{4}{n}\sum_{i=1}^{n} \mathbb{E}[\|\delta_i^{(k)}\|^2] + \frac{4}{n}\sum_{i=1}^{n} \mathbb{E}[\|\delta_i^{(k-1)}\|^2] \bigg)$$

$$\leq (1-\delta)^R \left( 4\mathbb{E}[\|\delta^{(k-1)}\|^2] + 4\mathbb{E}[\|\nabla f(x^{(k)})\|^2] \right.$$

$$+ \frac{4}{n}\sum_{i=1}^{n} \mathbb{E}[\|\delta_i^{(k)}\|^2] + \frac{4}{n}\sum_{i=1}^{n} \mathbb{E}[\|\delta_i^{(k-1)}\|^2] + \frac{4\sigma^2}{nR} \bigg). \tag{35}$$

For any $1 \leq i \leq n$, using the similar argument, we have that

$$\mathbb{E}[\|\delta_i^{(k)}\|^2] = \mathbb{E}[\|\tilde{g}^{(k)} - \text{FCC}(\tilde{g}_i^{(k)}, C_i, R)\|^2] \leq (1-\delta)^R \mathbb{E}[\|\tilde{g}_i^{(k)}\|^2]$$

$$= (1-\delta)^R \mathbb{E}[\|\hat{g}_i^{(k)} + \delta_i^{(k-1)}\|^2]$$

$$\leq (1-\delta)^R \left( 2\mathbb{E}[\|\hat{g}_i^{(k)}\|^2] + 2\mathbb{E}[\|\delta_i^{(k-1)}\|^2] \right)$$

$$= (1-\delta)^R \left( 2\mathbb{E}[\|\nabla f_i(x^{(k)})\|^2] + 2\mathbb{E}[\|\delta_i^{(k-1)}\|^2] + \frac{2\sigma^2}{R} \right). \tag{36}$$

Note that for any $1 \leq i \leq n$,

$$\mathbb{E}[\|\nabla f_i(x^{(k)})\|^2] = \mathbb{E}[\|\nabla f_i(x^{(k)}) - \nabla f(x^{(k)}) + \nabla f(x^{(k)})\|^2]$$

$$\leq 2\mathbb{E}[\|\nabla f_i(x^{(k)}) - \nabla f(x^{(k)})\|^2] + 2\mathbb{E}[\|\nabla f(x^{(k)})\|^2]. \tag{37}$$

Taking the average of (36) over all $1 \leq i \leq n$, and using (37) and Assumption 5, we further have that

$$\frac{1}{n}\sum_{i=1}^{n} \mathbb{E}[\|\delta_i^{(k)}\|^2]$$

$$= (1-\delta)^R \left( \frac{4}{n}\sum_{i=1}^{n} \mathbb{E}[\|\nabla f_i(x^{(k)}) - \nabla f(x^{(k)})\|^2] + 4\mathbb{E}[\|\nabla f(x^{(k)})\|^2] \right.$$

$$+ \frac{2}{n}\sum_{i=1}^{n} \mathbb{E}[\|\delta_i^{(k-1)}\|^2] + \frac{2\sigma^2}{R} \bigg)$$

$$\leq (1-\delta)^R \left( 4\mathbb{E}[\|\nabla f(x^{(k)})\|^2] + \frac{2}{n}\sum_{i=1}^{n} \mathbb{E}[\|\delta_i^{(k-1)}\|^2] + 4b^2 + \frac{2\sigma^2}{R} \right). \tag{38}$$

Summing up (35) and plugging (38) into it, we obtain

$$\mathbb{E}[\|\delta^{(k)}\|^2] + \frac{4}{n}\sum_{i=1}^{n} \mathbb{E}[\|\delta_i^{(k)}\|^2]$$

$$\leq (1-\delta)^R \left( 4\mathbb{E}[\|\delta^{(k-1)}\|^2] + 4\mathbb{E}[\|\nabla f(x^{(k)})\|^2] + \frac{4}{n}\sum_{i=1}^{n} \mathbb{E}[\|\delta_i^{(k)}\|^2] + \frac{4}{n}\sum_{i=1}^{n} \mathbb{E}[\|\delta_i^{(k-1)}\|^2] \right.$$

$$+ \frac{4\sigma^2}{nR} + 16\mathbb{E}[\|\nabla f(x^{(k)})\|^2] + \frac{8}{n}\sum_{i=1}^{n} \mathbb{E}[\|\delta_i^{(k-1)}\|^2] + 16b^2 + \frac{8\sigma^2}{R} \bigg)$$

$$\leq (1-\delta)^R \left( 4\mathbb{E}[\|\delta^{(k-1)}\|^2] + \frac{4}{n}\sum_{i=1}^{n} \mathbb{E}[\|\delta_i^{(k)}\|^2] + \frac{12}{n}\sum_{i=1}^{n} \mathbb{E}[\|\delta_i^{(k-1)}\|^2] \right.$$

$$+ 20\mathbb{E}[\|\nabla f(x^{(k)})\|^2] + 16b^2 + \frac{12\sigma^2}{R} \bigg).$$

We thus have

$$\mathbb{E}[\|\delta^{(k)}\|^2] + \frac{4(1-(1-\delta)^R)}{n} \sum_{i=1}^{n} \mathbb{E}[\|\delta_i^{(k)}\|^2]$$

$$\leq (1-\delta)^R \left( 4\mathbb{E}[\|\delta^{(k-1)}\|^2] + \frac{12}{n} \sum_{i=1}^{n} \mathbb{E}[\|\delta_i^{(k-1)}\|^2] + 20\mathbb{E}[\|\nabla f(x^{(k)})\|^2] + 16b^2 + \frac{12\sigma^2}{R} \right).$$

Since $R$ is sufficiently large such that

$$(1-\delta)^R < \frac{1}{4}, \tag{39}$$

we have that $4(1-(1-\delta)^R) \geq 3$ and hence

$$\mathbb{E}[\Psi^{(k)}] \leq \mathbb{E}[\|\delta^{(k)}\|^2] + \frac{4(1-(1-\delta)^R)}{n} \sum_{i=1}^{n} \mathbb{E}[\|\delta_i^{(k)}\|^2]$$

$$\leq 4(1-\delta)^R \left( \mathbb{E}[\Psi^{(k-1)}] + 5\mathbb{E}[\|\nabla f(x^{(k)})\|^2] + 4\left(b^2 + \frac{\sigma^2}{R}\right) \right).$$

$\square$

With Lemma 8, we easily reach its ergodic version:

**Lemma 9** (ERGODIC VANISHING ERROR). *Suppose $\theta \triangleq 4(1-\delta)^R < 1$. Then it holds that for any $K \geq 0$,*

$$\frac{1}{K+1} \sum_{k=0}^{K} \mathbb{E}[\Psi^{(k-1)}] \leq \frac{1}{K+1} \frac{5\theta}{1-\theta} \sum_{\ell=0}^{K-1} \mathbb{E}[\|\nabla f(x^{(\ell)})\|^2] + \frac{4\theta}{1-\theta} \left(b^2 + \frac{\sigma^2}{R}\right).$$

*Proof.* Let $\theta = 4(1-\delta)^R$, then by Lemma 8 and noting $\Psi^{(-1)} = 0$, we have

$$\mathbb{E}[\Psi^{(k)}] \leq \theta \left( \mathbb{E}[\Psi^{(k-1)}] + 5\mathbb{E}[\|\nabla f(x^{(k)})\|^2] + 4\left(b^2 + \frac{\sigma^2}{R}\right) \right)$$

$$\leq \theta \left( \theta \left( \mathbb{E}[\Psi^{(k-2)}] + 5\mathbb{E}[\|\nabla f(x^{(k-1)})\|^2] + 4\left(b^2 + \frac{\sigma^2}{R}\right) \right) \right.$$

$$\left. + 5\mathbb{E}[\|\nabla f(x^{(k)})\|^2] + 4\left(b^2 + \frac{\sigma^2}{R}\right) \right)$$

$$= \theta^2 \mathbb{E}[\Psi^{(k-2)}] + 5 \sum_{\ell=k-1}^{k} \theta^{k+1-\ell} \mathbb{E}[\|\nabla f(x^{(\ell)})\|^2] + 4 \sum_{\ell=k-1}^{k} \theta^{k+1-\ell} \left(b^2 + \frac{\sigma^2}{R}\right)$$

$$\leq \cdots$$

$$\leq \theta^{k+1} \mathbb{E}[\Psi^{(-1)}] + 5 \sum_{\ell=0}^{k} \theta^{k+1-\ell} \mathbb{E}[\|\nabla f(x^{(\ell)})\|^2] + 4 \sum_{\ell=0}^{k} \theta^{k+1-\ell} \left(b^2 + \frac{\sigma^2}{R}\right)$$

$$= 5 \sum_{\ell=0}^{k} \theta^{k+1-\ell} \mathbb{E}[\|\nabla f(x^{(\ell)})\|^2] + 4 \sum_{\ell=0}^{k} \theta^{k+1-\ell} \left(b^2 + \frac{\sigma^2}{R}\right). \tag{40}$$

Therefore, by taking the summation of (40) over $k = 0, \ldots, K-1$ and using $\Psi^{(-1)} = 0$ again, we further have

$$\frac{1}{K+1} \sum_{k=0}^{K} \mathbb{E}[\Psi^{(k-1)}] = \frac{1}{K+1} \sum_{k=0}^{K-1} \mathbb{E}[\Psi^{(k)}]$$

$$\leq \frac{1}{K+1} \sum_{k=0}^{K-1} \left( 5 \sum_{\ell=0}^{k} \theta^{k+1-\ell} \mathbb{E}[\|\nabla f(x^{(\ell)})\|^2] + 4 \sum_{\ell=0}^{k} \theta^{k+1-\ell} \left(b^2 + \frac{\sigma^2}{R}\right) \right)$$

$$= \frac{1}{K+1} \left( 5 \sum_{\ell=0}^{K-1} \mathbb{E}[\|\nabla f(x^{(\ell)})\|^2] \left( \sum_{k=\ell}^{K-1} \theta^{k+1-\ell} \right) + 4 \left(b^2 + \frac{\sigma^2}{R}\right) \sum_{\ell=0}^{K-1} \sum_{k=\ell}^{K-1} \theta^{k+1-\ell} \right),$$

where we change the summation order of indexes $k$ and $\ell$ in the last identity. Since $\sum_{k=\ell}^{K-1} \theta^{k+1-\ell} = \frac{\theta(1-\theta^{K-\ell})}{1-\theta} \leq \frac{\theta}{1-\theta}$, we thus have

$$\frac{1}{K+1} \sum_{k=0}^{K} \mathbb{E}[\Psi^{(k-1)}]$$

$$\leq \frac{1}{K+1} \left( 5 \sum_{\ell=0}^{K-1} \mathbb{E}[\|\nabla f(x^{(\ell)})\|^2] \frac{\theta}{1-\theta} + 4 \left(b^2 + \frac{\sigma^2}{R}\right) \frac{K\theta}{1-\theta} \right)$$

$$\leq \frac{1}{K+1} \frac{5\theta}{1-\theta} \sum_{\ell=0}^{K-1} \mathbb{E}[\|\nabla f(x^{(\ell)})\|^2] + \frac{4\theta}{1-\theta} \left(b^2 + \frac{\sigma^2}{R}\right).$$

$\square$

Given the above lemmas, now we prove the convergence rate of NEOLITHIC.

**Theorem 4.** *Let the communication round be* $R = \left\lceil \frac{\ln(n/\delta) + \ln(4\max\{b^2, \delta\sigma^2\})}{\delta} \right\rceil$ *and learning rate be as in* (46). *Then it holds that for any* $K \geq 0$,

$$\frac{1}{K+1} \sum_{k=0}^{K} \mathbb{E}[\|\nabla f(x^{(k)})\|^2] = \tilde{O}\left( \left(\frac{\Delta L \sigma^2}{nT}\right)^{\frac{1}{2}} + \frac{\Delta L}{\delta T} \right),$$

*where* $T = KR$ *is the total number of gradient queries (compressed communications) on each worker.*

*Proof.* Averaging (29) over $k = 0, \ldots, K$, and using the fact that $y^{(0)} = x^{(0)}$ and $f(y^{(K+1)}) \geq f^\star$, we have

$$\frac{1}{K+1} \sum_{k=0}^{K} \mathbb{E}[\|\nabla f(x^{(k)})\|^2]$$

$$\leq \frac{4\mathbb{E}[f(y^{(0)})] - 4\mathbb{E}[f(y^{(K+1)})]}{\gamma(K+1)} + \frac{8\gamma^2 L^2}{K+1} \sum_{k=0}^{K} \mathbb{E}[\|\Omega^{(k-1)}\|^2] + \frac{2\gamma L \sigma^2}{nR}$$

$$\leq \frac{4(f(x^{(0)}) - f^\star)}{\gamma(K+1)} + \frac{8\gamma^2 L^2}{K+1} \sum_{k=0}^{K} \mathbb{E}[\|\Omega^{(k-1)}\|^2] + \frac{2\gamma L \sigma^2}{nR}. \tag{41}$$

By the definition of $\Omega^{(k-1)}$ and using the Cauhy-Schwarz inequality, it holds that

$$\mathbb{E}[\|\Omega^{(k-1)}\|^2] = \mathbb{E}\left[ \left\| \delta^{(k-1)} + \frac{1}{n} \sum_{i=1}^{n} \delta_i^{(k-1)} \right\|^2 \right]$$

$$\leq \left(1 + \frac{1}{3}\right) \mathbb{E}[\|\delta^{(k-1)}\|^2] + (1+3)\mathbb{E}\left[ \left\| \frac{1}{n} \sum_{i=1}^{n} \delta_i^{(k-1)} \right\|^2 \right]$$

$$\leq \frac{4}{3} \mathbb{E}[\|\delta^{(k-1)}\|^2] + \frac{4}{n} \sum_{i=1}^{n} \mathbb{E}[\|\delta_i^{(k-1)}\|^2] = \frac{4}{3} \mathbb{E}[\Psi^{(k-1)}].$$

Therefore, by Lemma 9, we reach

$$\frac{1}{K+1} \sum_{k=0}^{K} \mathbb{E}[\|\Omega^{(k-1)}\|^2] \leq \frac{4}{3(K+1)} \sum_{k=0}^{K} \mathbb{E}[\|\Psi^{(k-1)}\|^2]$$

$$\leq \frac{20\theta}{3(K+1)(1-\theta)} \sum_{\ell=0}^{K-1} \mathbb{E}[\|\nabla f(x^{(\ell)})\|^2] + \frac{16\theta}{3(1-\theta)} \left(b^2 + \frac{\sigma^2}{R}\right). \tag{42}$$

Plugging (42) into (41), we reach that

$$\frac{1}{K+1} \sum_{k=0}^{K} \mathbb{E}[\|\nabla f(x^{(k)})\|^2]$$

$$\leq \frac{4(f(x^{(0)}) - f^\star)}{\gamma(K+1)} + \frac{160\gamma^2 L^2 \theta}{3(1-\theta)(K+1)} \sum_{\ell=0}^{K-1} \mathbb{E}[\|\nabla f(x^{(\ell)})\|^2]$$

$$+ \frac{128\gamma^2 L^2 \theta}{3(1-\theta)} \left(b^2 + \frac{\sigma^2}{R}\right) + \frac{2\gamma L \sigma^2}{nR}. \tag{43}$$

Assume the learning rate $\gamma$ is sufficiently small such that

$$\gamma \leq \frac{1}{11L} \sqrt{\frac{1-\theta}{\theta}} \quad \Longrightarrow \quad \frac{160\gamma^2 L^2 \theta}{3(1-\theta)} \leq \frac{1}{2}. \tag{44}$$

then we can further bound (43) as

$$\frac{1}{K+1} \sum_{k=0}^{K} \mathbb{E}[\|\nabla f(x^{(k)})\|^2] \leq \frac{8\Delta}{\gamma(K+1)} + \frac{4\gamma L \sigma^2}{nR} + \frac{256\gamma^2 L^2 \theta}{3(1-\theta)} \left(b^2 + \frac{\sigma^2}{R}\right). \tag{45}$$

where we use the definition $\Delta := f(x^{(0)}) - f^\star$. By choosing

$$\gamma = \frac{1}{11L + \sigma \left(\frac{(K+1)L}{2nR\Delta}\right)^{\frac{1}{2}} + \left(\frac{32(K+1)L^2\theta(b^2 + \sigma^2/R)}{3(1-\theta)\Delta}\right)^{\frac{1}{3}}}, \tag{46}$$

we have

$$\frac{1}{K+1} \sum_{k=0}^{K} \mathbb{E}[\|\nabla f(x^{(k)})\|^2]$$

$$\leq \frac{16\sigma\sqrt{L\Delta}}{\sqrt{nR(K+1)}} + \frac{36\Delta^{\frac{2}{3}} L^{\frac{2}{3}} \theta^{\frac{1}{3}} (b^2 + \sigma^2/R)^{\frac{1}{3}}}{(K+1)^{\frac{2}{3}}(1-\theta)^{\frac{1}{3}}} + \frac{88L\Delta}{K+1}$$

$$= O\left(\left(\frac{\Delta L \sigma^2}{nT}\right)^{\frac{1}{2}} + \frac{\theta^{\frac{1}{3}} \Delta^{\frac{2}{3}} L^{\frac{2}{3}} \max\{b^{\frac{2}{3}}, \sigma^{\frac{2}{3}}/R^{\frac{1}{3}}\}}{(1-\theta)^{\frac{1}{3}} K^{\frac{2}{3}}} + \frac{\Delta L}{K}\right) \tag{47}$$

where we bound $f(x^{(0)}) - f^\star$ by $\Delta$ in the last equation.

Let

$$R = \left\lceil \frac{\max\{\ln\left(\frac{\delta T \max\{b^2, \sigma^2\delta\}}{\Delta L}\right), \ln(8)\}}{\delta} \right\rceil = \tilde{O}\left(\frac{1}{\delta}\right), \tag{48}$$

then it holds that

$$\theta = 4(1-\delta)^R \leq 4e^{-\delta R}$$

$$\leq 4\exp\left(-\max\{\ln(\delta T \max\{b^2, \sigma^2\delta\}/\Delta L), \ln(8)\}\right) = \min\left\{\frac{4\Delta L}{\delta T \max\{b^2, \sigma^2\delta\}}, \frac{1}{2}\right\}, \tag{49}$$

and hence $1 - \theta = \Omega(1)$ and $\gamma$ satisfies (44). Plugging (48) and (49) into (47), and applying the notation $T = KR$, we reach

$$\frac{1}{K+1} \sum_{k=0}^{K} \mathbb{E}[\|\nabla f(x^{(k)})\|^2] = O\left(\left(\frac{\Delta L \sigma^2}{nT}\right)^{\frac{1}{2}} + \frac{\theta^{\frac{1}{3}} R^{\frac{2}{3}} \Delta^{\frac{2}{3}} L^{\frac{2}{3}} \max\{b^{\frac{2}{3}}, \sigma^{\frac{2}{3}}/R^{\frac{1}{3}}\}}{T^{\frac{2}{3}}} + \frac{R\Delta L}{T}\right)$$

$$= \tilde{O}\left(\left(\frac{\Delta L \sigma^2}{nT}\right)^{\frac{1}{2}} + \frac{\theta^{\frac{1}{3}} \Delta^{\frac{2}{3}} L^{\frac{2}{3}} \max\{b^{\frac{2}{3}}, \sigma^{\frac{2}{3}}\delta^{\frac{1}{3}}\}}{\delta^{\frac{2}{3}} T^{\frac{2}{3}}} + \frac{\Delta L}{\delta T}\right)$$

$$= \tilde{O}\left(\left(\frac{\Delta L \sigma^2}{nT}\right)^{\frac{1}{2}} + \frac{\Delta L}{\delta T}\right). \tag{50}$$

$\square$

## C Experiment Supplement

### C.1 Synthetic Dataset

**Linear regression.** We consider the following least-square problem:

$$\min_{x\in\mathbb{R}^d}\quad\frac{1}{2n}\sum_{i=1}^{n}\|A_ix-b_i\|^2.$$

Coefficient matrix $A_i$ and measurement $b_i$ are associated with node $i$, and $M$ is the size of local data. We set $d=30$, $n=32$ and $M=1000$, and generate data by letting each node $i$ be associated with a local solution $x_i^\star$ randomly generated by $\mathcal{N}(0,I_d)$. Then we generate each element in $A_i$ following standard normal distribution, and measurement $b_i$ is generated by $b_i=A_ix_i^\star+s_i$ with white noise $s_i\sim\mathcal{N}(0,0.01)$. At each query, every node will randomly sample a row in $A_i$ and the corresponding element in $b_i$ to evaluate the stochastic gradient. We adopt the rand-1 compressor and set the number of rounds $R=4$ for NEOLITHIC. We use stair-wise decaying learning rates in which the learning rates are divided by every $2,500$ communication rounds. Each algorithm is averaged with 20 trials. The result is shown in Figure 1 (left). It is observed that NEOLITHIC outperforms MEM-SGD and Double-Squeeze in convergence rate, and it performs closely to P-SGD.

**Logistic regression.** We consider the following logistic regression problem:

$$\min_{x\in\mathbb{R}^d}\frac{1}{n}\sum_{i=1}^{n}f_i(x)\quad\text{where}\quad f_i(x)=\frac{1}{M}\sum_{m=1}^{M}\ln\left(1+\exp\left(-y_{i,m}h_{i,m}^\top x\right)\right)$$

where $\{h_{i,m},y_{i,m}\}_{m=1}^{M}$ is the training dateset held by node $i$ in which $h_{i,m}\in\mathbb{R}^d$ with $d=30$ is a feature vector while $y_{i,m}\in\{-1,+1\}$ is the corresponding label. Similar to the least square, each node $i$ is associated with a local solution $x_i^\star$. We generate each feature vector $h_{i,m}\sim\mathcal{N}(0,I_d)$, and label $y_{i,m}=1$ with probability $1/(1+\exp(-y_{i,m}h_{i,m}^\top x_i^\star))$; otherwise $y_{i,m}=-1$. We adopt the rand-1 compressor and set the number of rounds $R=4$ for NEOLITHIC. We use stair-wise decaying learning rates in which the learning rates are divided by every $800$ communication rounds. Each algorithm is averaged with 20 trials. The result is shown in Figure 2. It is observed that NEOLITHIC outperforms MEM-SGD and Double-Squeeze in convergence rate, and it performs closely to P-SGD.

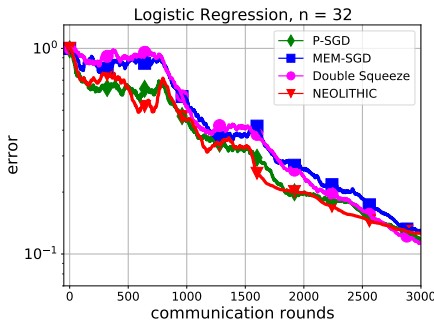

Figure 2: Convergence results on the synthetic logistic regression problem in terms of the mean-square error $\mathbb{E}[\|\nabla f(x)\|^2]$ versus communication rounds.

### C.2 Deep Learning Tasks

**Implementation details.** We implement all compression algorithms with PyTorch[47] 1.8.2 using NCCL 2.8.3 (CUDA 10.1) as the communication backend. For PSGD, we used PyTorch's native Distributed Data Parallel (DDP) module. All deep learning training scripts in this section run on a server with 8 NVIDIA V100 GPUs in our cluster and each GPU is treated as one worker.

**Image classification.** We investigate the performance of the aforementioned methods with CIFAR-10 [37] dataset. For CIFAR-10 dataset, it consists of 50,000 training images and 10,000 validation images categorized in 10 classes. We utilize two common variants of ResNet [27] model on CIFAR-10 (ResNet-20 with roughly 0.27M parameters and ResNet-18 with 11.17M parameters). We train

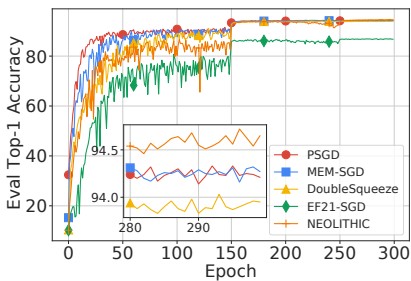

Figure 3: Convergence results on the CIFAR-10 in terms of validation accuracy.

Table 4: Accuracy comparison on CIFAR-10 with heterogeneous data (ResNet-20).

| METHODS | MEM-SGD | DOUBLE-SQUEEZE | EF21-SGD | NEOLITHIC | C-RATIO % |
|---|---|---|---|---|---|
| $\alpha = 1$ | $85.42 \pm 0.22$ | $83.72 \pm 0.17$ | $38.5 \pm 2.17$ | $\mathbf{85.47 \pm 0.12}$ | 1 |
| $\alpha = 10$ | $91.61 \pm 0.19$ | $91.25 \pm 0.17$ | $68.58 \pm 0.13$ | $\mathbf{91.76 \pm 0.13}$ | 1 |
| $\alpha = 1$ | $86.43 \pm 0.38$ | $86.13 \pm 0.21$ | $72.65 \pm 0.37$ | $\mathbf{87.17 \pm 0.24}$ | 5 |
| $\alpha = 10$ | $91.88 \pm 0.23$ | $91.66 \pm 0.11$ | $86.36 \pm 0.15$ | $\mathbf{92.14 \pm 0.22}$ | 5 |

total 300 epochs and set the batch size to 128 on every worker. The learning rate is set to 5e-3 for single worker and warmed up in the first 5 epochs and decayed by a factor of 10 at 150 and 250-th epoch. All experiments were repeated three times with different seeds. For NEOLITHIC, we set $R = 2$. Following previous works [62], we use top-k compressor with different compression ratio to evaluate the performance of the aforementioned methods. As shown in Figure 1 (left), Figure 3, and Table 2, NEOLITHIC consistently outperforms other compression methods and reach the similar performance to PSGD. It is worth noting that EF21-SGD, while guaranteed to converge with milder assumptions than ours, does not provide competitive performance in deep learning tasks listed in Tables 2, 3, and 4. We find our reported result for EF21-SGD is consistent with Fig. 7 (the left plot) in [24].

**Performance with heterogeneous data.** We simulate data heterogeneity among nodes via a Dirichlet distribution-based partitioning with parameter $\alpha$ controlling the data heterogeneity. The training data for a particular class tends to concentrate in a single node as $\alpha \rightarrow 0$, i.e. becoming more heterogeneous, while the homogeneous data distribution is achieved as $\alpha \rightarrow \infty$. We test $\alpha = 1$ and 10 in Table 4 for all compared methods as corresponding to a setting with high/low heterogeneity.

**Effects of accumulation rounds.** We also empirically evaluate the performance of NEOLITHIC with different choice of parameter $R$ in deep learning tasks. NEOLITHIC have slightly performance degradation in both compression scenarios as $R$ scales up. We conjecture that the gradient accumulation step, which amounts to using large-batch samples in gradient evaluation, can help in the optimization and training stage as proved in this paper, but it may hurt the generalization performance. We recommend using NEOLITHIC in applications that are friendly to large-batch training.

Table 5: Effects of round numbers for CIFAR-10 dataset with ResNet-18

| ROUNDS | 2 | 3 | 4 | 5 |
|---|---|---|---|---|
| NEOLITHIC (5%) | $94.63 \pm 0.09$ | $93.32 \pm 0.08$ | $92.55 \pm 0.12$ | $91.48 \pm 0.18$ |
| NEOLITHIC (1%) | $94.155 \pm 0.10$ | $93.15 \pm 0.11$ | $92.27 \pm 0.08$ | $91.32 \pm 0.12$ |

# D  More Details of Table 1

There exist mismatches between some rates listed in Table 1 and those established in literature. The mismatches exist because

1. we have strengthened the vanilla rates by relaxing their restrictive assumptions (say, Double-Squeeze) or uncovering the hidden terms (say, CSER);

2. we have extended the vanilla rates to the same setting as NEOLITHIC (say, extend MEM-SGD to non-convex and smooth setting, or transform QSGD to the distributed setting).

With these modifications, these baseline algorithms can be compared with NEOLITHIC in a fair manner. Next we clarify each modification one by one.

### D.1 Q-SGD

There is a slight inconsistency between the original rate in [32] and the one listed Table 1 due to the following reasons:

1. We noticed that the rate stated in [32, Corollary 3] is not optimal following [32, Theorem 2] since the authors set the learning rate as $\theta\sqrt{M/T}$ where $\theta$ is a universal constant. The rate in [32, Corollary 3] can be slightly improved in terms of $\sigma^2, b^2, \Delta = f(x^{(0)}) - \min_x f(x)$ by involving them into the learning rate. We manually optimize the learning rate by choosing $\Theta((L + (T(1+\omega)L\sigma^2/\Delta M)^{1/2} + (\omega LTb^2/n\Delta)^{1/2})^{-1})$.

2. $M$ is the total mini-batch size on all workers in Q-SGD (see the paragraph of contribution, page 2, [32]). For a fair comparison with other methods in Table 1, We set $M = n$ to make the number of total gradient queries per iteration equivalent in all algorithms.

### D.2 CSER

While [68, Corollary 1] does not have a $O(1/T)$ term in the convergence rate, it does impose another condition $T \gg n$ to the convergence statement. If $T \gg n$ holds, then $1/\sqrt{nT} \gg 1/T$ and hence the $1/T$ term is dominated by $1/\sqrt{nT}$ and thus hidden in the notation $O(\cdot)$. However, the condition does not appear in the convergence theorem of NEOLITHIC. To conduct a fair comparison, we have to remove condition $T \gg n$ from its convergence theorem, which thus incurs additional $O(1/T)$ term in the rate accordingly. The rate we provided in Table 1 is hence more precise than that in [68].

### D.3 Double-Squeeze

The original rate of Double-Squeeze established in [62] is based on an unrealistic assumption that accumulated compression errors are bounded by unknown $\epsilon$ (see [62, Assumption 1.3]). This makes its rate incomparable with other methods. However, we can remove the unrealistic assumption and easily derive a comparable bound where $\epsilon$ can be explicitly replaced with $O(G/\delta^2)$. We plug $O(G/\delta^2)$ into [62, Corollary 2] to get the rate listed in our Table 1.

To this end, we follow the notations of [62] to derive explicit upper bounds for server/worker compression errors $\mathbb{E}[\|\boldsymbol{\delta}_t\|]$ and $\mathbb{E}[\|\boldsymbol{\delta}_t^{(i)}\|]$ which, combined with [62, Corollary 2], leads to the rate in our Table 1. We consider compressors $Q$ (in server) and $Q_i$ (in worker $i$) utilized are $\delta$-contractive. We use $\boldsymbol{g}_t^{(i)}$ to indicate local (stochastic) gradient.

**Bound of local compression error:** $\mathbb{E}[\|\boldsymbol{\delta}_t^{(i)}\|] = O(G/\delta)$. By $\delta$-contraction and Young's inequality, we have for any $\rho > 0$ that

$$
\begin{aligned}
\mathbb{E}[\|\boldsymbol{\delta}_t^{(i)}\|^2] =& \mathbb{E}[\|\boldsymbol{v}_t^{(i)} - Q_i(\boldsymbol{v}_t^{(i)})\|^2] \\
\leq& (1-\delta)\mathbb{E}[\|\boldsymbol{v}_t^{(i)}\|^2] = (1-\delta)\mathbb{E}[\|\boldsymbol{g}_t^{(i)} + \boldsymbol{\delta}_{t-1}^{(i)}\|^2] \\
\leq& (1+\rho)(1-\delta)\mathbb{E}[\|\boldsymbol{g}_t^{(i)}\|^2] + (1+1/\rho)(1-\delta)\mathbb{E}[\|\boldsymbol{\delta}_{t-1}^{(i)}\|^2]
\end{aligned}
\tag{51}
$$

Iterating (51) for $t, t-1, \ldots, 0$ and noting $\boldsymbol{\delta}_0^{(i)} = 0$, we reach

$$
\begin{aligned}
\mathbb{E}[\|\boldsymbol{\delta}_t^{(i)}\|^2] \leq& (1+\rho)(1-\delta)\sum_{s=1}^t (1+1/\rho)^{t-s}(1-\delta)^{t-s}\mathbb{E}[\|\boldsymbol{g}_s^{(i)}\|^2] \\
\leq& \frac{(1+\rho)(1-\delta)G^2}{1-(1+1/\rho)(1-\delta)}
\end{aligned}
\tag{52}
$$

where the last inequality holds because $\mathbb{E}[\|\boldsymbol{g}_s^{(i)}\|^2] \leq G^2$ for all $1 \leq s \leq t$. Here one must choose $\rho = \Omega(1/\delta)$ to avoid the explosion of the upper bound, which leads to $\mathbb{E}[\|\boldsymbol{\delta}_t^{(i)}\|^2] = O(G^2/\delta^2)$. Therefore, by Jessen's inequality, we have $\mathbb{E}[\|\boldsymbol{\delta}_t^{(i)}\|] \leq \sqrt{\mathbb{E}[\|\boldsymbol{\delta}_t^{(i)}\|^2]} = O(G/\delta)$.

**Bound of global compression error:** $\mathbb{E}[\|\boldsymbol{\delta}_t\|] = O(G/\delta^2)$. By $\delta$-contraction and Young's inequality, we have for any $\rho > 0$ that

$$
\begin{aligned}
\mathbb{E}[\|\boldsymbol{\delta}_t\|^2] =& \mathbb{E}[\|\boldsymbol{v}_t - Q(\boldsymbol{v}_t)\|^2] \\
\leq& (1-\delta)\mathbb{E}[\|\boldsymbol{v}_t\|^2] = (1-\delta)\mathbb{E}\left[\left\|\frac{1}{n}\sum_{i=1}^n Q_i(\boldsymbol{v}_t^{(i)}) + \boldsymbol{\delta}_{t-1}\right\|^2\right] \\
\leq& (1-\delta)(1+\rho)\mathbb{E}\left[\left\|\frac{1}{n}\sum_{i=1}^n Q_i(\boldsymbol{v}_t^{(i)})\right\|^2\right] + (1+1/\rho)(1-\delta)\mathbb{E}[\|\boldsymbol{\delta}_{t-1}\|^2]. \quad (53)
\end{aligned}
$$

Again by Young's inequality and $\delta$-contraction, we have

$$
\begin{aligned}
\mathbb{E}\left[\left\|\frac{1}{n}\sum_{i=1}^n Q_i(\boldsymbol{v}_t^{(i)})\right\|^2\right] \leq& 2\mathbb{E}\left[\left\|\frac{1}{n}\sum_{i=1}^n Q_i(\boldsymbol{v}_t^{(i)}) - \boldsymbol{v}_t^{(i)}\right\|^2\right] + 2\mathbb{E}\left[\left\|\frac{1}{n}\sum_{i=1}^n \boldsymbol{v}_t^{(i)}\right\|^2\right] \\
\leq& \frac{2-\delta}{n}\sum_{i=1}^n \mathbb{E}[\|\boldsymbol{v}_t^{(i)}\|^2] = O(G^2/\delta^2) \quad (54)
\end{aligned}
$$

where the last identity is because the upper bound of $\mathbb{E}[\|\boldsymbol{v}_t^{(i)}\|^2]$ can be obtained by following the derivation in (51) and (52). Taking $\rho = 2(1-\delta)/\delta = O(1/\delta)$ in (53) and using (54), we reach an inequality taking a form like

$$
\mathbb{E}[\|\boldsymbol{\delta}_t\|^2] \leq (1-\delta/2)\mathbb{E}[\|\boldsymbol{\delta}_{t-1}\|^2] + O(G^2/\delta^3). \quad (55)
$$

Iterating (55) similarly, we easily reach $\mathbb{E}[\|\boldsymbol{\delta}_t\|^2] = O(G^2/\delta^4)$ and thus $\mathbb{E}[\|\boldsymbol{\delta}_t\|] = O(G/\delta^2)$.

### D.4  MEM-SGD

We notice that only the rate for strongly convex problems is established in the original MEM-SGD paper [58]. To compare it fairly with NEOLITHIC, we derive its convergence rate in the non-convex setting by ourselves.

The main recursion of MEM-SGD is

$$
\mathbf{p}_t^i = \eta\nabla f_i(\mathbf{x}_t, \xi_t^i) + \mathbf{e}_t^i, \quad \mathbf{x}_{t+1} = \mathbf{x}_t - \frac{1}{n}\sum_{i=1}^n Q_i(\mathbf{p}_t^i), \quad \mathbf{e}_{t+1}^i = \mathbf{p}_t^i - Q_i(\mathbf{p}_t^i).
$$

The key steps of our derivation are listed as follows.

1. Following the similar argument to (51) and (52), we can bound the compression error as $\mathbb{E}[\|\mathbf{e}_t^i\|^2] = O(\eta^2 G^2/\delta^2)$. Note that here $\eta^2$ appears since compression is conducted after multiplying the learning rate $\eta$.

2. The recursion formula of MEM-SGD is $\mathbf{y}_{t+1} = \mathbf{y}_t - \frac{\eta}{n}\sum_{i=1}^n \nabla f_i(\mathbf{x}_t, \xi_t^i)$ with $\mathbf{y}_t \triangleq \mathbf{x}_t - \frac{1}{n}\sum_{i=1}^n \mathbf{e}_t^i$. In fact, one can easily check that

$$
\begin{aligned}
\mathbf{x}_{t+1} =& \mathbf{x}_t - \frac{1}{n}\sum_{i=1}^n Q_i(\mathbf{p}_t^i) \\
=& \mathbf{x}_t - \frac{1}{n}\sum_{i=1}^n (\mathbf{p}_t^i - \mathbf{e}_{t+1}^i) = \mathbf{x}_t - \frac{1}{n}\sum_{i=1}^n (\eta\nabla f_i(\mathbf{x}_t, \xi_t^i) + \mathbf{e}_t^i - \mathbf{e}_{t+1}^i) \\
=& \mathbf{y}_t - \frac{\eta}{n}\sum_{i=1}^n \nabla f_i(\mathbf{x}_t, \xi_t^i) + \frac{1}{n}\sum_{i=1}^n \mathbf{e}_{t+1}^i.
\end{aligned}
$$

Table 6: Comparison between between Q-SGD [32], VR-MARINA [63], and SASHA-MVR [26]. To explicitly clarify the influence of different compression strategies, we keep the stochastic gradient variance $\sigma^2$, data heterogeneity bound $b^2$, mini-batch size $B$ (used in [32, 26]), probability of conducting uncompressed communication $p$ (used in [63]), but omit smoothness constant $L$, and initialization gap $f(x^{(0)}) - f^\star$ in the below results.

| Algorithm | #Communication | #Gradient Query per Worker |
|---|---|---|
| Q-SGD [32] | $O\left(\frac{1}{nB\epsilon^4}\left((1+\omega)\sigma^2 + \omega b^2\right)\right)$ | $B\times$#Communication |
| VR-MARINA [26] | $O\left(\frac{1}{\epsilon^2}\left(1+\sqrt{\frac{1-p}{pn}\left(\omega + \frac{1+\omega}{\max\{1,\frac{\sigma^2}{n\epsilon^2}\}}\right)}\right)\right)$ | $\max\{1,\frac{\sigma^2}{n\epsilon^2}\}\times$#Communication |
| DASHA-MVR [63] | $O\left(\frac{1}{\epsilon^2}\left(1+\frac{\sigma}{n\epsilon B^{3/2}} + \frac{\sigma^2}{\epsilon^2 B}\right)\right)$ | $B\times$#Communication |

3. Following the derivation of (34), one can obtain

$$\mathbb{E}[f(\mathbf{y}_{t+1})] - \mathbb{E}[f(\mathbf{y}_t)] \leq 2\eta L^2 \mathbb{E}[\|\mathbf{y}_t - \mathbf{x}_t\|^2] - \frac{\eta(1-\eta L)}{2}\mathbb{E}[\|\nabla f(\mathbf{x}_t)\|^2] + \frac{\eta^2 L\sigma^2}{2n}. \tag{56}$$

Setting $\eta \leq \frac{1}{2L}$ such that $\frac{\eta(1-\eta L)}{2} \geq \frac{\eta}{4}$ and rearranging (56), we have

$$\mathbb{E}[\|\nabla f(\mathbf{x}_t)\|^2] \leq \frac{4(\mathbb{E}[f(\mathbf{y}_t)] - \mathbb{E}[f(\mathbf{y}_{t+1})])}{\eta} + \frac{2\eta L\sigma^2}{n} + 8L^2\mathbb{E}[\|\mathbf{y}_t - \mathbf{x}_t\|^2]. \tag{57}$$

4. By the definition of $\mathbf{y}_t$ and step 1., we have

$$\mathbb{E}[\|\mathbf{y}_t - \mathbf{x}_t\|^2] \leq \frac{1}{n}\sum_{i=1}^n \mathbb{E}[\|\mathbf{e}_t^i\|^2] = O(\eta^2 G^2/\delta^2). \tag{58}$$

Averaging (57) with (58) plugged into, we reach

$$\frac{1}{T}\sum_{t=0}^{T-1}\mathbb{E}[\|\nabla f(\mathbf{y}_t)\|^2] \leq O\left(\frac{\mathbb{E}[f(\mathbf{x}_0)] - f^\star}{\eta} + \frac{\eta L\sigma^2}{n} + \frac{\eta^2 L^2 G^2}{\delta^2}\right). \tag{59}$$

Setting the learning rate $\eta = (2L + (\frac{L\sigma^2}{n\Delta})^{1/2} + (\frac{L^2 G^2}{\delta^2\Delta})^{1/3})^{-1}$ in (59) leads to the rate we listed in Table 1.

## D.5 Comparison with More Algorithms

We supplement the comparison between Q-SGD [32], VR-MARINA [26], and DASHA-MVR [63] in Table 6. The results are compared in terms of the communication/query complexity to reach $\mathbb{E}[\|\nabla f(x)\|^2] \leq \epsilon$ for a sufficiently small $\epsilon$.

Several additional comments are as follows:

1. All three algorithms utilize unidirectional, unbiased, and independent compressors.

2. All algorithms conduct an imbalanced number of compressed communications and of gradient queries. We therefore list the communication and gradient query complexity separately in Table 6. The result of Q-SGD is slightly tuned by us, see the argument in Appendix D.1.

3. MARINA (with $O(1/\epsilon^2)$) and SASHA (with $O(1/\epsilon^3)$) have better communication complexity than QSGD (with $O(1/\epsilon^4)$) when $\epsilon$ is sufficiently small.