# OpenReview forum: "Lower Bounds and Nearly Optimal Algorithms in Distributed Learning with Communication Compression"
_NeurIPS.cc/2022/Conference — NeurIPS 2022 Accept_

### Official Review · Reviewer_nEJw · 2022-07-06

**Rating:** 6
**Confidence:** 4
**Soundness:** 3 good
**Presentation:** 3 good
**Contribution:** 3 good

**Summary:**

This work considers smooth and non-convex distributed learning problem under communication compression schemes to reduce communication costs between the server and devices. It discusses widely adopted classes of unbiased and contractive compression operators, unidirectional (device to server) and bidirectional (device to server and server to device) communication compression schemes. The paper proposes lower bounds on the number of stochastic gradient evaluations (or communication rounds) for all these regimes and designs a method, called NEOLITHIC, that achieves those lower bounds for the mentioned regimes up to log factors.

**Questions:**

Please see the comments under Strengths And Weaknesses section. The critical points are detailed comments (1) and (2).

**Ethics Review Area:**

["I don’t know"]

**Limitations:**

IMO, no need.

**Strengths And Weaknesses:**

**(1: Lower bounds).** There are 4 lower bounds discussed in the paper: two classes of compression operators (unbiased or contractive) and two types of communication reduction schemes (unidirectional or bidirectional). The first result (Theorem 1) shows the lower bound for unidirectional unbiased compression and the other three lower bounds are corollaries from Theorem 1 following standard arguments (unidirectional compression is a special case of bidirectional one and unbiased compression can be treated as a special case of contractive compression). ***However, the lower bound (3) of Theorem 1 does not seem consistent with existing upper bounds.*** As you mention in lines 64-65, algorithms with unidirectional and unbiased compressors enjoy better convergence rates. Specifically, convergence rates with unbiased compressions typically depends on $\frac{\omega}{n}$. For example, the method MARINA from [27](Gorbunov et. al. 2021 - Marina: faster non-convex distributed learning with compression) has $(1+\frac{\omega}{n})\frac{1}{T}$ rate with full gradients (i.e., $\sigma=0$). On the other hand, the lower bound (3) claims $(1+\omega)\frac{1}{T}$ rate when $\sigma=0$. In other words, the upper bound from [27] is better than the proposed lower bound (3). As the other lower bounds are special cases of (3), ***the correctness of the lower bounds are questionable.***


**(2: Upper bound).** ***The proof of the convergence rate of the proposed NEOLITHIC algorithm has an issue.*** The proof of Theorem 3 goes quite neat and clean up to equation (45). To get to the issue, consider (44) showing the following rate for the method (ignoring constants):
$$
\mathbf{E} [ || \nabla f(\hat{x}) ||^2 ] \lesssim \frac{1}{\gamma K} + \frac{\gamma \sigma^2}{n R} + \gamma^2\theta.
$$

Then, choosing the stepsize as in (45), it continues (before equation (46))
$$
\mathbf{E} [ || \nabla f(\hat{x}) ||^2 ] \lesssim \frac{1}{\gamma K} + \frac{\gamma \sigma^2}{n R} + \gamma^2\theta  {\color{red} \lesssim \frac{\sigma}{\sqrt{n R K}} + \frac{\theta^{1/3}}{K^{2/3}} + \frac{1}{K}}.
$$

***I do not think that this upper bound in red holds.*** In particular, $\frac{1}{\gamma K}$ term ($\frac{8\Delta}{\gamma(K+1)}$ exact expression) in (44) is upper bounded by $\frac{1}{K}$ (exact expression is $\frac{88L\Delta}{K+1}$) using the upper bound $\gamma \le \frac{1}{11L}$ from (45). However, since $\gamma$ is in the denominator of $\frac{1}{\gamma K}$ term, we have the ***reverse inequality***, namely $\frac{8\Delta}{\gamma(K+1)} {\color{red}\ge} \frac{88L\Delta}{K+1}$. In order to upper bound $\frac{1}{\gamma K}$ by $\frac{1}{K}$, one need to have a lower bound for $\gamma$. But, the choice (45) for $\gamma$ does not allow a lower bound, namely $\gamma\to0$ as $T\to\infty$.


Moreover, ***I do not think this issue can be fixed and the same rate of Theorem 3 be proven***. In order to minimize the sum of first two terms $\frac{1}{\gamma K} + \frac{\gamma \sigma^2}{n R}$, one needs to choose $\gamma$ such that $\frac{1}{\gamma K} = \frac{\gamma \sigma^2}{n R}$, which implies that $\gamma = \frac{1}{\sigma}\sqrt{\frac{nR}{K}}$. With this choice of $\gamma$, the first two terms recover the desired term in the rate, namely $\frac{1}{\gamma K} + \frac{\gamma \sigma^2}{n R} \lesssim \frac{\sigma}{\sqrt{nT}}$. However, the third term $\gamma^2\theta$ becomes $\frac{nR}{K\sigma^2}\frac{1}{\delta T} \lesssim \frac{n}{\sigma^2} \frac{1}{\delta^3 T^2}$, which blows up when $\sigma\approx 0$.

--------

**(3: References on Table 1).** Does the reference [32] really showed the mentioned rate, which blows up for $\sigma\approx 0$ ? MEM-SGD paper [53] seems to be on convex problems, could you point the place in [53] where the non-convex rate is mentioned ? There are some typos in the rate of DoubleSqueeze, I think the rate is $\mathcal{O}(\frac{\sigma}{\sqrt{nT}} + \frac{G^{2/3}}{\delta^{1/3}T^{2/3}} + \frac{1}{T})$.

**(4: On FCC).** I understand why FCC (with $R>1$) is introduced and used in the proof. However, FCC sends $R$ times more bits and I think this cost outweighs the exponential decay of the error shown in Lemma 2. For example, applying Top-$2$ is (slightly) better applying Top-$1$ twice (namely FCC with Top-$1$ and $R=2$). The reason is that theoretical contraction factor of Top-$2$ is smaller than of FCC with $R=2$, i.e., $1-\frac{2}{d} < (1-\frac{1}{d})^2$.

**(5: On Remark 2).** Check the MCM algorithm of (Philippenko and Dieuleveut 2021 - Preserved central model for faster bidirectional compression in distributed settings) related to bidirectional/unidirectional communication compression.

---

> ### Author Response · Authors · 2022-08-02
> **Response to Reviewer nEJw (Part II)**
>
> \
> **3. References on Table 1.**
>
> \
> **Q-SGD**: We apologize for the typo in Q-SGD's rate. It should be $\frac{(1+\omega)^{0.5}\sigma+\omega^{0.5}b}{\sqrt{nK}}$. We will fix it in the revision.
>
> **MEM-SGD**: We agree that only strongly-convex rate is established in the original MEM-SGD paper [53]. Since MEM-SGD is a milestone communication compression algorithm with effective practical performance, we list it as an important baseline. To compare it fairly with NEOLITHIC, we derive its convergence rate in the non-convex setting by ourselves. Since this derivation is not the focus of our paper, we did not discuss it in detail in the paper. To address the reviewer's concerns, we provide an anonymous material to list several key steps in the derivation, see Sec. 2 in [R1].
>
> **DoubleSqueeze**: We believe the result listed in Table 1 should be correct. The original rate of DoubleSqueeze established in [57] is based on an unrealistic assumption that accumulated compression errors are bounded by unknown $\epsilon$  (see Assumption 1.3, [57]). This makes its rate incomparable with other methods. However, we can remove the unrealistic assumption and easily derive a comparable bound where $\epsilon$ can be explicitly replaced with $O(G/\delta^2)$. We plug $O(G/\delta^2)$ into Corollary 2, [57] to get the rate listed in our Table 1. We provide an anonymous material [R1] which lists the key steps to derive the upper bounds of compression errors, see Sec. 1 in [R1].
>
> \
> [R1] An anonymous material provided by Paper8019 authors, https://anonymous.4open.science/r/NeurIPS2022-Paper8019/paper8019_more_details.pdf. (Please download it for a better view)
>
> \
> **4. On FCC**
>
> \
> We thank the reviewer for bringing up this interesting discussion.
>
> - For **specific** compressors such as top-k, we agree with that there can be smarter ways than FCC, e.g., enlarging compression ratio in one-shot to balance the compression error and cost.
>
> - However, we remark that FCC, as **a universal algorithmic tool**, is a **general strategy** that can apply to a broad class of compressors such as quantization whose compression/transmission ability may not be adjusted proportionally. In other words, given any compressor satisfying Assumption 3 or 4, one can always utilize FCC strategy to achieve the optimal convergence rate.
>
> - Moreover, we have theoretically shown that FCC can enable algorithms with optimal convergence rate. Therefore, we emphasize the advantages of FCC on its role in matching the optimal rate.
>
> - Finally, as suggested by reviewer wUMh, a similar nested structure also exists in literature. This means the FCC-type structure has applications in compression scenarios.
>
> \
> **5. On Remark 2**
>
> \
> Thanks for providing this related paper. For **unbiased compressors and (strongly) convex objectives**, the MCM algorithm of (Philippenko and Dieuleveut) does perform bidirectional compression and achieves the same convergence as algorithms using unidirectional compression. We will cite and carefully discuss this closely-related paper in the revision to properly adjust our Remark 2.
>
> \
> **6. Summary**
>
> \
> We hope the rebuttal can clarify the reviewer's confusions. Since **no mistakes exist in the established lower and upper bounds**, we ask the reviewer to kindly re-evaluate our work. We are looking forward to further follow-up discussions, and more than happy to clarify any further comments.

---

> ### Author Response · Authors · 2022-08-02
> **Response to Reviewer nEJw (Part I)**
>
> \
> Many thanks for the careful review. However, some comments from the reviewer **have factual mistakes**. We have attempted to clarify all the questions as best as we can. We are glad to address any further questions or comments.
>
> \
> **1. Lower bounds.**
>
> \
> **Our lower bounds do not contradict existing upper bounds**. The reason why [27] provides a smaller upper bound is it makes a stronger assumption that "the compressor utilized in local workers are **independent to each other**". In contrast, as we stated in line 168 in the main text, our lower bound metric **does not require the compressors to be independent**. It applies to a general class of compressors that can be dependent or independent to each other. In other words, [27] essentially considers a smaller and sub-class of compressors (equipped with independence) and hence is able to achieve an smaller upper bound.
>
> \
> Can [27] removes the independence assumption while maintaining its upper bound? No it cannot. In the first equation in page 19, [27] established
> $$\mathbb{E}\left[\left\Vert \frac{1}{n}\sum_{i=1}^n (Q\_i(z_i)-z_i)\right\Vert^2\right]=\frac{1}{n^2}\sum_{i=1}^n\mathbb{E}\left[\left\Vert Q\_i(z_i)-z_i\right\Vert^2\right]$$ where $Q_i$ is the local compressor and $z_i=\nabla f_i(x^{k+1})-\nabla f_i(x^k)$ is a local variable related to the algorithm details. This equation holds only **when the local compressors are unbiased and independent**. Without independence, one can only get
> $$\mathbb{E}\left[\left\Vert \frac{1}{n}\sum_{i=1}^n (Q\_i(z_i)-z_i)\right\Vert^2\right]\leq \frac{1}{n}\sum_{i=1}^n\mathbb{E}\left[\left\Vert Q\_i(z_i)-z_i\right\Vert^2\right]$$ by using the Cauchy-Schwartz inequality. Note that there can be gap of factor $n$. That is why [27] shows the rate $(1+\frac{\omega}{n})\frac{1}{T}$ when $\sigma=0$. **Without the independence,  [27] cannot show a better rate than our lower bound**. We believe the inconsistency the reviewer found with unbiased compressors can also be traced back to this tricky setup difference.
>
> \
> Finally, we would like to emphasize again that it is not strange that rates can be improved with stronger assumptions. For example, it is well-known that the convergence rate of strongly-convex functions is faster than that of generally-convex (but not strongly-convex) functions. We hope this analogy can further resolve the reviewer's confusion.
>
> \
> **2. Upper bound.**
>
> \
> The proof of the convergence rate of NEOLITHIC algorithm **does not have an issue.** We now argue why the inequality in red holds. Consider the universal form $\mathbb{E}[\Vert \nabla f(x)\Vert^2]\leq \frac{A}{\gamma K}+\frac{B\gamma}{ nR}+C(b^2+\sigma^2/R)\gamma^2$. Here, we use $A$, $B$, and $C$ to denote quantities that do not relate to $n$, $K$, and $R$ for clarity. Now let us take
> $$\gamma =\frac{1}{11L +(\frac{KB}{AnR})^\frac{1}{2} +(\frac{KC(b^2+\sigma^2/R)}{A})^\frac{1}{3}},$$
> then term $\frac{B\gamma}{ nR}$ and term $C(b^2+\sigma^2/R)\gamma^2$ can be easily upper bounded by $(\frac{AB}{nKR})^\frac{1}{2}$ and ${A^\frac{2}{3}C^\frac{1}{3}(b^2+\sigma^2/R)^\frac{1}{3}}/{K^\frac{2}{3}}$ since $\gamma \leq (\frac{AnR}{KB})^\frac{1}{2} $ and $\gamma \leq (\frac{A}{KC(b^2+\sigma^2/R)})^\frac{1}{3} $. For term $\frac{A}{\gamma K}$, by the definition of $\gamma$, we have
> $$\frac{A}{\gamma K}= \frac{A}{K}\left( 11L +(\frac{KB}{AnR})^\frac{1}{2} +(\frac{KC(b^2+\sigma^2/R)}{A})^\frac{1}{3} \right)\\
> =\frac{11AL}{K}+(\frac{AB}{nKR})^\frac{1}{2}+\frac{A^\frac{2}{3}C^\frac{1}{3}(b^2+\sigma^2/R)^\frac{1}{3}}{K^\frac{2}{3}}.$$
> Note that the last two quantities are also the upper bounds of term $\frac{B\gamma}{ nR}$ and term $C(b^2+\sigma^2/R)\gamma^2$.
> Summarizing the argument above, we reach
> $$\frac{A}{\gamma K}+\frac{B\gamma}{ nR}+C(b^2+\sigma^2/R)\gamma^2\leq \frac{11AL}{K}+2(\frac{AB}{nKR})^\frac{1}{2}+2\frac{A^\frac{2}{3}C^\frac{1}{3}(b^2+\sigma^2/R)^\frac{1}{3}}{K^\frac{2}{3}}$$
> where the last two quantities are doubled to provide a joint upper bound for the summation.
>
> \
> That said, we are not bound $\frac{A}{\gamma K}$ individually by $O(\frac{AL}{K})$ as the reviewer did. Instead, we directly substitute the exact value of $\gamma$ to $\frac{A}{\gamma K}$ without any inequality.
>
> \
> These arguments clearly shows that our proof is **correct**. We hope our clarification can remove the reviewer's confusion. We are more than happy for further discussion.

---

> ### Author Response · Authors · 2022-08-06
> **Could you please confirm whether our response have addressed your questions?**
>
> \
> Dear Reviewer nEJw,
>
> \
> Thanks very much for your careful review. Our paper **does not have the mistakes** you mentioned in the lower and upper bound. Could you please confirm whether our response have addressed your questions? Can you kindly re-evaluate our paper? We are happy to address any further questions or comments.
>
> \
> Best,\
> Paper8019 Authors

---

> > ### Comment · Reviewer_nEJw · 2022-08-07
> > **Thank you for the clarifications!**
> >
> > Dear Authors,
> >
> > **Thank you for the clarifications! My two main concerns on lower and upper bounds are addressed. I will increase my score shortly!**
> >
> >
> > ----
> >
> >
> >
> > Please also comment on these two follow-up questions.
> >
> > **1.** Now I see that the setup you consider for the lower bound is different than I thought: your minimax measure in (2) has $\sup$ over compressors and not $\inf$. The other two $\sup$ parts (one for the functional class and one for the oracle) are reasonable to take $\sup$ as we are not in full control of those aspects of the training (collected data influences the loss function and oracle via subsampling). On the other hand, we take $\inf$ over the algorithm class as we have full control over what algorithm to use during the training. I think the same should be for the compressors (namely $\inf$ term) as we fully control what compressor to use during the training.
> >
> > > Based on this observation above, what is the importance of the considered setup (minimax measure (2)) for the lower bound? NEOLITHIC achieves that lower bound, but you can exploit the compression operator (just like the optimization algorithm) and obtain better rates.
> >
> > I understand the analogy you mentioned for strongly convex and convex cases. But loss functions depend on training data in some way that we are not in full control of; hence we consider the worst-case scenario (i.e., $\sup$ over functional class).
> >
> >
> > **2.** Agreed, FCC is a general tool. However, do you have any example of a compressor for which performing FCC with $R>1$ is better than single compression with an enlarged ratio? I am not convinced that big $R$ should be any better than small $R$. As you also pointed out in the rebuttal, $R=2$ gives the best performance.

---

> > > ### Author Response · Authors · 2022-08-08
> > > **Further Response (Part II)**
> > >
> > > \
> > > **3. Toward the optimal compressor**
> > >
> > > \
> > > While “optimal compressor” is not the purpose of this paper, we are happy to communicate with the reviewers about this topic.
> > >
> > > 3.1. If only the compressor conditions specified in this paper are considered, that is, $\mathcal{U}\_\omega$ (unbiased) and $\mathcal{C}_\delta$ (contractive), which have been extensively used in the existing literature to model and analyze compressors, the identity operator, the uncompressed compressor, is the optimal compressor. This is because these conditions specify the minimum performance of the compressors, instead of their maximum performance.
> > >
> > > 3.2 Mathematically, defining and finding the "optimal" compressor is still an open problem. As the arguments in 2.1 reveal, establishing "an optimal compressor" requires a new and meaningful setting beyond the two classes: $\mathcal{U}\_\omega$ and $\mathcal{ C}_\delta$. To the best of our knowledge, research in this area is rather limited.
> > >
> > >
> > > \
> > > **4. Big R is not necessarily better**
> > >
> > > \
> > > We did not claim in the paper that larger R$ is better in the paper or in our reply. "R = 2 for optimal performance" merely summarizes the question raised by reviewer MLpo.
> > >
> > > \
> > > Let us clarify our experimental results. They mainly show test accuracy rather than theoretically predicted optimization performance. They do not indicate that a larger $R$ is always better. The best choice of $R$ depends on the heterogeneity $b^2$, the noise level $\sigma^2$, and other properties. The noise level $\sigma^2$ is related to our chosen minibatch size $128$.
> > > We believe that changing $128$ to other values will lead to other "best values" for $R$. One evidence is our use of $R=4$ in the synthetic experiments of linear regression and logistic regression.

---

> > > ### Author Response · Authors · 2022-08-08
> > > **Further Response (Part I)**
> > >
> > > \
> > > Thanks for the valuable comments. Our response is as follows. We are happy to clarify any further questions.
> > >
> > > \
> > > **1. Why "sup over compressor" instead of "inf over compressor"**
> > >
> > > \
> > > "inf over compressor" aims to find the best compressor, which may seem appropriate, but we have a different objective in this paper, which requires us to use "sup over compressor."
> > >
> > > \
> > > There have been many recent works proposing new compressors for various distributed problem settings. But, almost all compressor performance analyses come down to one of the only two properties: unbiasedness or contraction. This leads to a natural question:
> > >
> > > >If we want to improve the convergence rate of distributed optimization with communication compression, should we continue to use those properties and focus on how to use them more cleverly in distributed algorithms (for example, develop a more effective error feedback strategy), or should we look for new compressor properties?
> > >
> > > \
> > > To answer the question above, we need to know the **theoretical limits** imposed by those two properties. If the realizable performance is still far from the limit, then maybe we haven't used the current compressors properly, so we should focus on clever combinations of distributed algorithms and current compressors; if the limit is almost reached (which is the conclusion demonstrated in this paper through "sup over compressor" and our development of NEOLITHIC), then we have to find a new compressor property (note: our conclusion is not to say that existing compressors are bad, but their compression analyses are close to the limit). So, if one wants to establish a better compressor performance than the lower bounds of this paper, they must discover a fundamentally new compressor property out of existing or new compressors, say the independence property.
> > >
> > > \
> > > **2. FCC compressor**
> > >
> > > \
> > > Our FCC approach becomes natural once you understand what our "sup over compressor" lower bounds express. Instead of arguing that FCC is a great compressor, we use it to show that our theoretical lower bounds are tight and achievable; otherwise, the reader may wonder if the proposed lower bounds are meaningful enough. Note that we must use something that works with ALL compressors satisfying one of those two properties. A nice but non-general technique will fail to prove the tightness of our lower bounds.
> > >
> > > \
> > > Since FCC indeed helps us obtain the lower bounds (up to a log factor), it is a very good general-purpose technique. This fact is not contradicted when a specific compressor can replace FCC in a specific setting and works better.

---

### Official Review · Reviewer_MLpo · 2022-07-11

**Rating:** 7
**Confidence:** 4
**Soundness:** 4 excellent
**Presentation:** 4 excellent
**Contribution:** 3 good

**Summary:**

This paper is focused on the compression technique of distributed ML. It considers two well-studied types of compressors, i.e., the unbiased compressor and the contractive compressor. The paper provides lower bounds for those compression algorithms in the context of both unidirectional and bidirectional communication. Based on this theory, the authors come up with a new distributed training algorithm called NEOLITHIC. According to the experiment results, NEOLITHIC can achieve better training accuracy compared with the existing compressed algorithm.

**Questions:**

- As mentioned in the appendix, it seems that 2 is the best value for communication round R. Can you explain why?
- Assuming R=2, then NEOLITHIC has 2 times more communication overhead than other methods, therefore it only has T/2 iterations for training. But with half number of iterations, how can you guarantee the best accuracy? In other words, in order to achieve the best accuracy, do you still need to train with NEOLITHIC for the same iterations as other methods?
- Given your theory, can you give compression guidelines for a distributed task? For example, for a given task (optimization problem), how can you decide the following: 1) which compressor can achieve the best accuracy 2) how much is the compression ratio 3) unidirectional or bidirectional?

**Limitations:**

Please refer to the weakness.

**Strengths And Weaknesses:**

- Strength
    - This paper takes unbiased and contractive compressors, unidirectional and bidirectional communication patterns all into a unified analysis, which can be quite helpful to better understand this field from a new perspective.
    - The theoretical analysis is solid and clear.
    - The proposed algorithm NEOLITHIC is quite creative in the sense that it involves multiple communication rounds to send a vector once.
- weakness
    - In the experiment, the authors only use contractive compressors, like random-1, top-k, but not unbiased compressors. So the effectiveness of quantization is not clear

---

> ### Author Response · Authors · 2022-08-02
> **Response to Reviewer MLpo (Part II)**
>
>
> \
> **3. We train NEOLITHIC for $T/2$ iterations**
>
> \
> When $R=2$, NEOLITHIC only runs $T/2$ iterations. However, NEOLITHIC samples $R$ times more data points in each iteration than other compression methods (listed in our Table 1) due to gradient accumulation. For example, if mini-batch size is set as $128$, NEOLITHIC will sample $128\times R$ data in each worker per iteration. That said, while NEOLITHIC only runs $T/2$ iterations in the training stage, it has traversed through the same number of data samples as the other methods which do not utilize gradient accumulation but run $T$ iterations. This is the main reason why NEOLITHIC can perform comparably well with the other approaches in $T/2$ iterations.
>
> \
> **4. Compression guidelines**
>
> \
> Our theory provides theoretical guarantees that bidirectional compression can perform as well as unidirectional compression.  As a result, we recommend using bidirectional compression in practice.
>
> \
> However, while our theory establishes an optimal convergence rate for the family of contractive or unbiased compressor. It does not clarify what compressor achieves best performance, and what the optimal ratio is. These two question are orthogonal to our theory, and we unfortunately cannot answer it at this moment.
>
> \
> But we are glad to share some empirical experience on compressors.
>
> - First, use compression only when it is necessary. If the communication overhead is not the concern, we suggest not using compression as it may hurt test accuracy.
>
> - Second, since the the distribution of the model and the gradient varies drastically between different tasks and optimizers, it is not possible to find a compressor that performs uniformly well for all tasks.
>
> - Third, for vision tasks, we recommend top-k compressor with a compression ratio between 0.1% to 5%.
>
> - Forth, we can compare the convergence between normal (no compression conducted) and compression training for a few iterations and choose the proper compression setting.  The early-stopping-like strategy, which works empirically well in tuning learning rate, can be also utilized in adjusting compression setting.
>
> \
> **5. Please be not affected by Reviewer nEJw**
>
> \
> The comments by Reviewer nEJw **have factual mistakes**. We have clarified all of them crystally clear in his response.
>
> \
> We hope the above explanation can clarify the reviewer's questions. We are looking forward to further follow-up discussions with the reviewer, and more than happy to clarify any further comments.

---

> > ### Comment · Reviewer_MLpo · 2022-08-08
> > **Response to the authors**
> >
> > Thanks for your reply! It has addressed my concerns, and I don't have more questions.

---

> ### Author Response · Authors · 2022-08-02
> **Response to Reviewer MLpo (Part I)**
>
> \
> We thank the reviewer for the positive comments and valuable questions. We have attempted to clarify all the questions as best as we can. We are glad to address any further comments or questions.
>
> \
> **1. NEOLITHIC with quantization**
>
> \
> As requested by the reviewer, we have conducted a set of new experiments utilizing quantization compressors. In the experiments, we used a 4-bit quantization compressor as described in QSGD [R1]. The other setting follows Table 2&3 in the paper. While we tuned very hard in EF21-SGD, it cannot achieve comparable performance to the other methods. We thus did not list EF21-SGD in the table.
>
> \
> It is observed that NEOLITHIC can achieve a slightly better test accuracy (especially in ResNet18) than other baselines.
>
> | Methods      | ResNet18 | ResNet20 |
> | ----------- | ----------- | ----------- |
> | PSGD      | 93.99 ±0.52     |     91.62 ±0.13    |
> | QSGD      | 92.86 ±0.34     |     90.24 ±0.22    |
> | MEM-SGD      | 93.47 ±0.27     |     91.36 ±0.07    |
> | Double-Squeeze      | 93.35 ±0.39     |     90.89 ±0.14   |
> | NEOLITHIC      | 93.87 ±0.46     |     91.35 ±0.14    |
>
> [R1] Alistarh, Dan, et al.  QSGD: Communication-Efficient SGD via Gradient Quantization and Encoding. NeurIPS 2017
>
>
> \
> **2. $R=2$ enables best performance**
>
> \
> We fully understand the reviewer's confusion. It is shown by theory that a certain vale for $R$ can enable NEOLITHIC with optimal convergence rate, but why does $R=2$ show the best empirical performance?  The reason lies in the mismatch between training loss and the generalization performance.
>
> \
> Note that our theory predicts the performance of NEOLITHIC in terms of the gradient norm of training losses, see the metric in Eq. (2). It therefore focuses more on the training performance of NEOLITHIC, rather than the generalization performance in terms of the test accuracy. It is well recognized in the deep learning community that a small training error may not necessarily indicate good generalization performance due to overfitting. That's why we tune $R$ manually in real experiments.
>
> \
> We empirically evaluate the performance of $R$ in table 5, which shows $R=2$ provides the best performance. Note that in addition to utilizing the multiple-round FCC compressor, NEOLITHIC also uses gradient accumulation $\hat{g}\_i = \frac{1}{R}\sum_{r=0}^{R-1} O\_i(x; \xi\_i^{(k,r)})$ per iteration to balance gradient queries with communication rounds. In other words, NEOLITHIC is essentially utilizing a large batch of samples per iteration.  It is known in the community that large-batch can help training, but may hurt the generalization, which we believe is the main reason why we cannot set $R$ too large.
>
> \
> Does $R=2$ always provide best empirical performance? We believe it does not. If NEOLITHIC is used in applications that are friendly to large-batch training, we can use a larger $R$. Also, the value of $R$ is also correlated with the mini-batch size we are using. In our current experiments, we used $128$ as the mini-batch size which means NEOLITHIC sampled $128\times R$ data in each worker per iteration. We believe if mini-batch size is switched to other number, the empirically optimal choice of $R$ may vary.

---

### Official Review · Reviewer_wUMh · 2022-07-11

**Rating:** 8
**Confidence:** 3
**Soundness:** 4 excellent
**Presentation:** 3 good
**Contribution:** 4 excellent

**Summary:**

The paper studies the problem of minimizing the objective, defined as a  finite sum of smooth and possibly non-convex functions in a Federated learning setting.
This paper aims to tackle the problem of expensive communications that appeared in the distributed scenario between one central server and multiple nodes\workers.
In particular, this work considers unidirectional and bidirectional communication compression as the approach to tackle the problem of communication bottleneck.
Looking into details authors establish the following list of contributions:

1) They establish convergence lower bounds for distributed algorithms with communication compression in the stochastic non-convex regime. This lower bound is applicable to any algorithm performing unidirectional or bidirectional compression and using unbiased or contractive compressors. They clearly compare the established lower bounds and convergence rates of the existing methods.

2) They propose a novel, nearly optimal algorithm with compression called NEOLITHIC that provably matches the established lower bound. Moreover, NEOLITHIC can adopt either unidirectional or bidirectional compression and is compatible with both unbiased and contractive compressors.

3) In particular, the convergence result of NEOLITHIC shows that algorithms using biased contractive compressors bidirectionally can theoretically converge as fast as those with unbiased compressors used unidirectionally.

4) In addition, they provided a set of numerical experiments comparing their novel method with existing baselines.


**Questions:**

1) Experiments on least squares and logistic regression:
Regarding compression properties, it is only said that the rand-1 compressor is utilized.
- 1) Could you please clarify whether you are using scaled rank-1% in all methods? Do you use unidirectional or bidirectional compression for NEOLITHIC?
- 2) Is there any reason why you exclude EF21-SGD from the comparison?
- 3) What stepsizes did you use? Were they constant or decreasing?

2) Could you please compare rates of VR-MARINA and DAHSA with QSGD that was mentioned in Table 1?

3) Could you please clarify where did you get the following rate for MEM-SGD (mentioned in Table 1):

$\mathcal O\left(\frac{\sigma}{\sqrt{n T}}+\frac{G^{2 / 3}}{\delta^{2 / 3} T^{2 / 3}}+\frac{1}{T}\right)$

I checked the original work [4] and does not have any results on non-convex losses. In fact, it deals only with convex losses.

4) The same question about method CSER [5].
I checked the original work, and it shows (Corollary 1 on page 20) that the rate of CSER does not have a term with 1/T dependence. Could you please clarify this inconsistency with Table 1?

5) I checked the original paper on Double Sqeeze, and their rate is

$\frac{1}{T} \sum_{t=0}^{T-1} \mathbb E \lVert\nabla f\left(\boldsymbol{x}_{t}\right)\rVert^{2} \lesssim \frac{\sigma}{\sqrt{n T}}+\frac{\epsilon^{\frac{2}{3}}}{T^{\frac{2}{3}}}+\frac{1}{T}$
where $\epsilon$ is the noise bound, such that

$ \mathbb E_{\omega} \lVert {\delta}_{t}^{(i)} \rVert \leq \frac{\epsilon}{2},  \forall t, \forall i $

$ \mathbb E_{\omega} \lVert {\delta}_{t} \rVert \leq \frac{\epsilon}{2},  \forall t$.

However, Table 1 says that Double Squeeze has a rate

$\mathcal O\left(\frac{1}{\sqrt{n T}}+\frac{G^{2 / 3}}{\delta^{4 / 3} T^{2 / 3}}+\frac{1}{T}\right)$

(the second term differs compared to the original work).

Could you please clarify this inconsistency?

6) Regarding QSGD, using the notation of Paper8019, Corollary 3 in [6] states the rate

$\mathcal O\left(\frac{(1 + \omega)\sigma^2 + \omega b^2}{\sqrt{MT} } \right)$, where M is minibatch size (not a number of workers $n$).

At the same time, Table 1 claims

$\mathcal O\left(\frac{(1+\omega) \sigma+\omega b^{2} / \sigma}{\sqrt{n T}}\right)$.

Could you please clarify this inconsistency in the QSGD rate from Table 1 and the original result from [6]?

References:

[4] S. U. Stich, J.-B. Cordonnier, and M. Jaggi. Sparsified sgd with memory. In Advances in Neural 445
Information Processing Systems, 2018.

[5] C. Xie, S. Zheng, O. Koyejo, I. Gupta, M. Li, and H. Lin. Cser: Communication-efficient sgd 464
with error reset. In Advances in Neural Information Processing Systems, 2020.

[6] Jiang, Peng, and Gagan Agrawal. "A linear speedup analysis of distributed deep learning with sparse and quantized communication." Advances in Neural Information Processing Systems 31 (2018).

**Limitations:**

1) I would recommend making the y-axis in the right subfigure of Figure 1 logarithmic scale. Otherwise, it is hard to distinguish plots corresponding to different methods.

2) One more possibly relevant and missing citation is [7]. The Algorithm 3PCv3 (Appendix C.6, page 26) already employs a similar nested structure as proposed in Paper8019 by FCC.

3) I would recommend running least squares and logistic regression experiments for a longer period. It looks like the methods on the left subfigure of Figure 1 and Figure 2 were stopped quite early and did not reach the SGD-specific oscillation region.

4) Some minor notes:
- 1) (line 685): instead of Cauchy-Schwarz, one needs to refer to Young's inequality for product;
- 2) (line 693): instead of Cauchy-Schwarz, one needs to refer to Jensen's inequality

FINAL REMARKS:

I would be happy to rate this paper an 8 for its solid theoretical contributions and reliable experiments.

However, at this moment, I can not do so since the paper still contains several crucial issues that are needed to be clarified or fixed.

I am ready to reconsider my current rate during rebuttals once you respond to me on Weaknesses 2-4 and Questions 1 - 6.

UPDATE: After the Authors-Reviewers discussion, I decided to increase the score since my concerns were resolved.

References:
[7] Richtárik, Peter, Igor Sokolov, Ilyas Fatkhullin, Elnur Gasanov, Zhize Li, and Eduard Gorbunov. 2022. “3PC: Three Point Compressors for Communication-Efficient Distributed Training and a Better Theory for Lazy Aggregation.” arXiv [cs.LG]. arXiv. http://arxiv.org/abs/2202.00998.

**Strengths And Weaknesses:**

Strengths:

1) The paper is very well written and easily read; the main claims are outlined. The paper adequately mentions related works, including other papers establishing lower bounds. In addition, it clearly explains
their relationship to this one. All notations that is being used in the theoretical claims are rigorously defined.

2) To the best of my knowledge, this is the first work establishing lower bounds for stochastic methods involving unidirectional and bidirectional compressions in a Federated learning setting. It closes an essential gap in such a relevant area.

3) Being a solid theoretical paper, it provides an extensive amount of experiments. Firstly, they confirm that NEOLITIC has a similar convergence rate as parallel-SGD, MEM-SGD, and Double Squeeze on least squares and logistic regression problems. Secondly, On the CIFAR10 classification problem via ResNet, train loss of NEOLITHIC converges slightly better than Double Squeeze and EF21-SGD, and it is compatible with MEM-SGD. However, the Top1 accuracy metric of NEOLITHIC outperforms all considered baselines in experiments on ResNet18 and ResNet20 with 1% and 5% compression ratios.

Weaknesses:

1)
The main theoretical flaw is that the analysis of NEOLITHIC relies on a restrictive Assumption 5 (bounded dissimilarity):

$$\frac{1}{n} \sum_{i=1}^{n} \lVert \nabla f_{i}(x)-\nabla f(x) \rVert^{2} \leq b^{2}, \quad \forall x \in \mathbb R^{d}$$

One can easily come up with an example for which this assumption does not generally hold.
For example, let us consider $f_{i}(x)=x^{\top} \mathbf A_{i} x$, where $\mathbf A_{i} \in \mathbb R^{d \times d}$. Since $\nabla f_{i}(x)=\mathbf B_{i} x$, where $\mathbf B_{i}=\mathbf A_{i}+\mathbf A_{i}^{\top}$.  The bounded dissimilarity assumption (Assumption 5), which can be written in the form $\frac{1}{n} \sum_{i=1}^{n}\left\|\left(\mathbf B_{i}-\frac{1}{n} \sum_{j=1}^{n} \mathbf B_{j}\right) x\right\|^{2} \leq b^{2}$, also does not hold, unless $\mathbf B_{i}=\mathbf B_{j}$ for all $i, j$, which reduces to the identical data regime, which is of limited interest.

2) Some details on the experimental setting are missing. See Question 1) in the next section.

3) Literature review ignores several papers that are seemed to be relevant [1], [2]. It seems VR-MARINA for online problems from [1] and DASHA-MVR from [2] both satisfy Assumption 2 and have a better rate than QSGD in the stochastic regime.  See Question 2) in the next section.

4) Table 1 contains possible typos:
- 1) It mentions a paper on MEM-SGD that does not have a non-convex rate. Their rate is applicable to strongly convex functions. I would recommend here to mention another, more relevant work [3].  See Question 3) in the next section.
- 2) Similar problems with CSER, Double Squeeze, and QSGD.  See corresponding Questions 4), 5) and 6) in the next section.

References:

[1] Gorbunov, Eduard, Konstantin Burlachenko, Zhize Li, and Peter Richtárik. 2021. “MARINA: Faster Non-Convex Distributed Learning with Compression.” arXiv [cs.LG]. arXiv. http://arxiv.org/abs/2102.07845.

[2] Tyurin, Alexander, and Peter Richtárik. 2022. “DASHA: Distributed Nonconvex Optimization with Communication Compression, Optimal Oracle Complexity, and No Client Synchronization.” arXiv [cs.LG]. arXiv. http://arxiv.org/abs/2202.01268.

[3] Koloskova, A., Lin, T., Stich, S. U., \& Jaggi, M.. Decentralized deep learning with arbitrary communication compression. arXiv preprint arXiv:1907.09356, 2019

---

> ### Author Response · Authors · 2022-08-02
> **Response to Reviewer wUMh (Part II)**
>
> \
> **4. Clarification for Table 1 (together with Questions 3, 4, 5, 6).**
>
> \
> Many thanks for bringing up so many valuable questions. There does exist mismatches between the rates listed in Table 1 and those established in literature. The mismatches exist because
>
> - we have strengthened the vanilla rates by relaxing their restrictive assumptions (say, DoubleSqueeze) or uncovering the hidden terms (say, CSER);
>
> - we have extended the vanilla rates to the same setting as NEOLITHIC (say, extend MEM-SGD to non-convex and smooth setting, or transform QSGD to the distributed setting).
>
> With these modifications, these baseline algorithms can be compared with NEOLITHIC in a fair manner. Next we clarify each modification one by one.
>
> \
> **CSER.** While Corollary 1 of [62] does not have a $1/T$ term in the convergence rate, it does impose another condition $T\gg n$ to the convergence statement. If $T\gg n$ holds, then $\frac{1}{\sqrt{nT}}\gg \frac{1}{T}$ and hence the ${1}/{T}$ term is dominated by $\frac{1}{\sqrt{nT}}$ and thus hidden in the $O(\cdot)$ notation. However, the condition $T\gg n$ does not appear in the convergence theorem of NEOLITHIC. To conduct a fair comparison, we have to remove condition $T\gg n$ from CSER convergence theorem, which thus incurs additional $1/T$ term in the rate accordingly. The existence of $1/T$ can be verified as follows. Recall the first term in Theorem 1 of [62] as $O(\frac{1}{\eta T})$. When $T$ is small (which does not satisfy $T \gg n$), it holds $\eta = 1/L$ according to the learning rate condition in Corollary 1. Substituting $\eta = 1/L$ to $O(\frac{1}{\eta T})$, we achieve $O(L/T)$. In other words, the dependence of $\frac{1}{T}$ is essential and cannot be erased from the rate in general. The rate we provided in Table 1 is hence more precise than that in [62].
>
> \
> **DoubleSqueeze.**  The original rate of DoubleSqueeze established in [57] is based on an unrealistic assumption that accumulated compression errors are bounded by unknown $\epsilon$ (see Assumption 1.3, [57]).  This makes its rate incomparable with other methods. However, we can remove the unrealistic assumption and easily derive a comparable bound where $\epsilon$ can be explicitly replaced with $O(G/\delta^2)$. We plug $O(G/\delta^2)$ into Corollary 2, [57] to get the rate listed in our Table 1.
> We provide an anonymous material [R1] which lists the key steps to derive the upper bounds of compression errors, see Sec. 1 in [R1].
>
> \
>  [R1] An anonymous material provided by Paper8019 authors, https://anonymous.4open.science/r/NeurIPS2022-Paper8019/paper8019_more_details.pdf.
>  (Please download it for a better view)
>
> \
> **Q-SGD.** This inconsistency was due to the following reasons:
>
> 1. We noticed that the rate stated in Corollary 3 [32] may not be optimal following its Theorem 2 since the authors set the step size as $\theta\sqrt{M/K}$ where $\theta$ is a universal real number. The rate in Corollary 3 can be slightly improved in terms of $\sigma^2$, $b^2$, $\Delta = f(x^0) -f^\star$ by involving them in the step size.
>
> 2. Our intention was to manually optimize the step size of [32] following its Theorem 2. However, we do recognize that we had an typo in the numerator $(1+\omega)\sigma+\omega b^2/\sigma$. It should be $(1+\omega)^{0.5}\sigma+\omega^{0.5}b$ where $L$ and $\Delta$ are hidden by setting $\gamma= \Theta\left(\left(L+(K(1+\omega)L\sigma^2/\Delta M)^{0.5}+(\omega L K b^2/n\Delta)^{0.5}\right)^{-1}\right)$ .
>
> 3. As the reviewer said, $M$ is the total mini-batch size on all workers in QSGD (see the paragraph of contribution, page 2, [32]). For a fair comparison with other methods in Table1, we set $M=n$ to make the number of total gradient queries per iteration equivalent in each algorithm. In fact, one can simply imagine that the minibatch stochastic gradient is computed by $n$ workers and that the minibatch size is $M = n$, see Sec. 2.2 in [R2].
>
> \
> [R2] Ji Liu and Ce Zhang, "Distributed Learning Systems with First-Order Methods", 2020.
>
> \
> **MEM-SGD**: We agree that only strongly-convex rate is established in the original MEM-SGD paper [53]. Since MEM-SGD is a milestone communication compression algorithm with effective practical performance, we list it as an important baseline. To compare it fairly with NEOLITHIC, we derive its convergence rate in the non-convex setting by ourselves. Since this derivation is not the focus of our paper, we did not discuss it in detail in the paper. To address the reviewer's concerns, we provide an anonymous material to list several key steps in the derivation, see Sec. 2 in [R1].

---

> > ### Author Response · Authors · 2022-08-02
> > **Response to Reviewer wUMh (Part II, continued)**
> >
> > \
> > **5. Response to Limitations.**
> >
> > \
> > We will re-plot the figure in the logarithmic scale in the revision.
> >
> > \
> > Many thanks for providing a useful reference. We will cite and discuss it carefully in the revision.
> >
> > \
> > Figures with longer periods are given in [R1]. In the newly provided figure, it is observed that all methods (except for EF21-SGD) converge to the SGD-specific oscillation region after sufficiently many iterations.
> >
> > \
> > Thanks very much for the minor notes. We will address them in the revision.
> >
> > \
> > **6. Summary**
> >
> > \
> > We thank the reviewer again for his careful and valuable comments. As requested by the reviewer, we have clarified all comments or questions on Assumption 5, experiments, comparison between MARINA, SASHA, and QSGD, the mismatch in Table 1, as well as other comments in Limitation. We hope these response can clarify the reviewer's questions.
> >
> > \
> > We are looking forward to the follow-up discussion with the reviewer, and more than happy to address any further comments or questions.

---

> > > ### Comment · Reviewer_wUMh · 2022-08-09
> > > **Conclusion**
> > >
> > > Dear Authors,
> > >
> > > Many thanks for your detailed and comprehensive answers.
> > > Indeed, they clarified my initial concerns.
> > > Please, consider adding these clarification details (as well as additional plots) to the revised version of the paper.
> > > Since concerns are resolved, I decided to increase my score from 6 to 8 to acknowledge the important contributions proposed in this paper.

---

> > > > ### Author Response · Authors · 2022-08-09
> > > > **Many thanks**
> > > >
> > > > \
> > > > Dear reviewer wUMh,
> > > >
> > > > \
> > > > We really appreciate your acknowledgement of our clarifications. We will definitely include these clarification details and the additional plots in our later revision.
> > > >
> > > > \
> > > > Best,
> > > >
> > > > The authors of paper 8019

---

> ### Author Response · Authors · 2022-08-02
> **Response to Reviewer wUMh (Part I)**
>
> \
> We thank the reviewer for the positive and detailed comments. All questions have been clarified as best as we can. We are glad to address any further comments or questions.
>
> \
> **1. Weakness in Assumption 5.**
>
> \
> We agree that Assumption 5 is restrictive in some examples. However, we remark that we can **replace Assumption 5 with a weaker one** (which is named as Assumption 5$^\prime$), i.e., $\frac{1}{n}\sum\_{i=1}^n\Vert \nabla f\_i(x)\Vert ^2\leq b^2+a^2\Vert \nabla f(x)\Vert ^2$ in which $a\geq 1$. Assumption 5 is a special case of Assumption 5$^\prime$ when $a=1$.
>
> \
> For the quadratic example provided by the reviewer, Assumption 5$^\prime$ is equivalent to $\frac{1}{n}\sum\_{i=1}^n\Vert  \mathbf{B}\_i  x \Vert^2 \leq b^2 + a^2\Vert \frac{1}{n}\sum_{i=1}^n \mathbf{B}\_i x \Vert^2$. Then it is guaranteed when $\frac{1}{n}\sum\_{i=1}^n  \mathbf{B}\_i^\top  \mathbf{B}\_i \preceq a^2(\frac{1}{n}\sum_{i=1}^n \mathbf{B}\_i)^T (\frac{1}{n}\sum_{i=1}^n \mathbf{B}\_i)$. In other words, Assumption 5$^\prime$ for quadratic objectives would not enforce homogeneity, i.e. $\mathbf{B}\_i=\mathbf{B}\_j$.
>
> \
> The current proof can be adapted to this new assumption without too much efforts. By slightly tweaking the step size and hyper-parameter $R$, we can obtain the same convergence rate for NEOLITHIC under Assumption 5$^\prime$  (logarithmic terms may change). To avoid confusing other reviewers, we do not make this change at this stage. But we can provide the proofs under this new assumption if the reviewer requests.
>
> \
> Lastly, we would like to emphasize that the current Assumption 5 is also standard in literature, see, e.g., [Assumption 1.3, 39] and [Assumption 5, 21]. It is milder than those made in many works such as [53, 57, 62] in which the bounded gradients are assumed. To our best knowledge, only EF21-SGD  is guaranteed to converge without Assumption 5, which, however, leads to a fairly loose convergence rate that cannot show linear-speedup.
>
> \
> **2. Experimental details (together with Question 1).**
>
> \
> In the experiments of least squares and logistic regression, we used rand-1 compressors for all algorithms except for P-SGD. Here rand-1 means we would randomly keep $1$ entry of the input vector and zero out the rest.  Note that rand-1 without scaling is contractive.
> We used bidirectional compression for NEOLITHIC.
>
> \
> We found EF21-SGD does not perform well in linear and logistic regressions, which is consistent with the experimental results listed in Tables 2 and 3. That's why we did not put it in simulations. We provide an anonymous material [R1] in which we compared the performance of EF21-SGD with other methods, see Sec 3 therein.
>
> \
> We used stair-wise decaying step-sizes in which the step sizes are divided by $2$ every 2500 (800) communication rounds in linear regression (logistic regression).
>
> \
> [R1] An anonymous material provided by authors, https://anonymous.4open.science/r/NeurIPS2022-Paper8019/paper8019_more_details.pdf.
> (Download it for a better view)
>
> \
> **3. VR-MARINA  and SASHA-MVR  (together with Question 2).**
>
> \
> We thank the reviewer for providing two useful papers. We will cite and discuss them carefully in the revision. A comparison between Q-SGD, VR-MARINA,  and SASHA-MVR is listed in the table below. The results are compared in terms of the communication/query complexity to reach $\mathbb{E}[\Vert \nabla f(x)\Vert^2]\leq \epsilon^2$ for a sufficiently small $\epsilon$.
>
> |  Algorithm   | #Communication  | #Gradient Query per Worker |
> |  ----  | ----  |  ----  |
> | Q-SGD  | $O\left(\frac{1}{Bn \epsilon^4}\left((1+\omega) \sigma^2+\omega b^2\right)\right)$ | $B \times \\#\text{Communication}$ |
> | VR-MARINA   | $O\left(\frac{\Delta L}{\epsilon^2}\left(1+\sqrt{\frac{1-p}{pn}\left(\omega +\frac{(1+\omega)}{\max(1,{\sigma^2}/{n\epsilon^2})}\right)}\right)\right)$ | $\Theta(\max(1,\frac{\sigma^2}{n\epsilon^2})) \times \\#\text{Communication}$ |
> | SASHA-MVR  | $O\left(\frac{\Delta}{\epsilon^2}\left(L\left(1+\frac{\sigma}{n\epsilon B^{1.5}}\right)\right)+\frac{\sigma^2}{\epsilon^2B}\right)$ | $B\times \\#\text{Communication}$ |
>
> In the table, $B$ is the mini-batch size utilized by each worker per iteration, and $p$ is the probability to conduct **uncompressed communication** in VR-MARINA, which, based on our understanding, follows a different algorithmic setup from ours.
>
> \
> Several additional comments are as follows:
>
> 1. All three algorithms utilize unidirectional, unbiased, and independent compressors.
>
> 2. Each algorithms conduct an imbalanced number of compressed communications and of gradient queries when stops. We therefore list the communication and gradient query complexity separately in the table. The results for Q-SGD is slightly tuned by us (see the argument towards "Q-SGD inconsistency" below).
>
> 3. MARINA (with $O(1/\epsilon^2)$) and SASHA (with $O(1/\epsilon^3)$) have better communication complexity than QSGD (with $1/\epsilon^4$) in theory.

---

> ### Author Response · Authors · 2022-08-09
> **Need more clarifications?**
>
> \
> Dear reviewer wUMh,
>
> \
> We thank you for your valuable comments and providing related references. We have made detailed responses to address your concerns, but we have not received your replies to our current clarifications yet. As the stage of author-reviewer discussion is about to end, we thus kindly ask if our responses have addressed all your concerns. If not, we are more than happy to provide more clarifications.
>
> \
> Best,
>
> The authors of paper 8019

---

### Author Response · Authors · 2022-08-06
**Can we have your response to our rebuttals?**

\
Dear reviewers,

\
Thank you very much for your careful review and valuable feedback. We have addressed all your questions. Could you please respond to our rebuttals? We are happy to clarify any further comments and questions.

\
Best,
Paper8019 Authors

---

### Meta-Review · Area_Chair_7wQc · 2022-08-20

**Recommendation:** Accept
**Confidence:** Certain

**Metareview:**

The authors have addressed many if not most of the issues raised, as evidenced by the increase in scores. All reviewers agree that the paper is worthy of acceptance. The analysis is insightful. The authors are suggested to contextualized with other gradient quantization schemes such as Gandikota et al, 2021, or lower bounds Mayekar et al., 2020; and also to motivate the "sup over compressors" measure.

**Award:**

No

---

### Decision · Program_Chairs · 2022-09-14

Accept